# The Price of Justice in Machine Learning: Fair Division with Subjective Value under Bounded Rationality

## Abstract

Statistical fairness criteria such as Demographic Parity, Equalized Odds, and Calibration are widely used in machine learning, but rate constraints do not by themselves specify how decision burdens are distributed across individuals. We develop a harm-allocation framework that models heterogeneous subjective error costs and evaluates decisions through fair-division axioms. Within this framework, we identify conditions under which common statistical criteria serve as valid surrogates for harm-based fairness, construct counterexamples where cost heterogeneity breaks this surrogate relationship, and reinterpret classic incompatibilities as conflicts between allocation principles. We further establish approximation lower bounds under bounded-information settings, including noisy or coarse cost information and restricted policy classes. Experiments connect these theoretical results to empirical behavior, showing when surrogate metrics align with or diverge from harm-based fairness, revealing approximation floors, and tracing these patterns across tasks, models, and fairness interventions under finer-grained harm diagnostics.

## 1 Introduction

Machine-learning (ML) systems increasingly mediate access to consequential opportunities, including credit, employment, insurance, education, and public services. A large body of work has therefore proposed *statistical* fairness criteria such as Demographic Parity (DP), Equalized Odds (EO), and Calibration, together with algorithms that enforce these criteria during training or through post-processing. These criteria are valuable tools for auditing, comparison, and intervention design. Their normative role, however, depends on what they are taken to represent: a constraint on observable rates, or a proxy for how the burdens of decisions are distributed among individuals.

This paper studies that relationship through a harm-allocation framework. In many deployment settings, a decision rule allocates both *goods*, such as approvals, access, and opportunities, and *bads*, such as false positives, false negatives, additional scrutiny, delay, stigma, and opportunity costs. The same statistical event can impose different burdens on different people. A false negative may be highly consequential for one applicant and less consequential for another; a false positive may create downstream risks that vary across social, institutional, and economic contexts. We capture this variation through heterogeneous subjective error costs and model each policy as inducing a vector of individual expected harms.

This perspective connects statistical fairness to the language of *fair division*. Given a policy-induced harm allocation, one can ask whether the resulting distribution of burdens satisfies axioms such as envy-freeness and proportionality-style protection. These axioms do not replace statistical fairness criteria. They provide a harm-based reference point for evaluating when criteria such as DP, EO, and Calibration function as valid surrogates. The resulting analysis is also complementary to similarity-based individual fairness: instead of asking whether similar individuals receive similar treatment, we ask how error burdens are distributed when individuals attach different costs to different decision errors.

A second component of the framework concerns institutional implementation. Fairness interventions are rarely chosen by an omniscient planner with direct access to individual costs and unrestricted policy freedom. They are implemented by institutions that operate with noisy or coarse cost information and with limited

policy classes, such as threshold rules or group-specific post-processing rules. We formalize this setting as bounded rationality and analyze the approximation floor that remains once informational and policy constraints are imposed.

The central question is therefore when low-dimensional statistical constraints can represent high-dimensional harm-allocation requirements. We characterize representation conditions under which statistical criteria align with harm-based axioms, construct heterogeneous-cost settings where the surrogate relationship breaks down, and reinterpret classic incompatibility results as conflicts among allocation principles. We then examine these claims empirically through theorem-guided experiments that trace surrogate–axiom alignment under cost heterogeneity, informational constraints, and different intervention families.

**Contributions.**

- **Harm allocation under subjective value.** Individual expected harm is formalized through heterogeneous error costs, so that each decision policy induces a population-level allocation of burdens rather than only a set of group-level rates.

- **Fair-division criteria for decision burdens.** Envy-freeness and proportionality-style protection are used to evaluate these harm allocations, providing an axiomatic reference point for comparing how ML decision burdens are distributed across individuals.

- **Surrogate validity and its limits.** The analysis identifies when DP, EO, and Calibration can represent harm-based fairness, and constructs heterogeneous-cost settings where rate-based criteria and harm-allocation criteria diverge.

- **Incompatibility as allocation conflict.** Classic impossibility and incompatibility results are reinterpreted as conflicts among allocation principles defined over policy-induced harms, clarifying the normative content of these trade-offs.

- **Bounded rationality and approximation floors.** Noisy or coarse cost information and restricted policy classes are incorporated into the framework, yielding lower bounds on the achievable violation of harm-based axioms.

- **Theorem-guided empirical analysis.** Experiments on ACS prediction tasks examine surrogate–axiom alignment across models and fairness interventions, with additional diagnostics for cost sensitivity, proxy validity, approximation floors, and harm-profile variation.

## 2 Related Work

### 2.1 Statistical Fairness as Surrogate Constraints

A central line of fair machine learning defines fairness through observable relations among predictions, labels, scores, and sensitive attributes. Early work distinguished individual and group notions of fairness and introduced statistical formulations of parity, representation, and similarity-based treatment (Dwork et al., 2012; Feldman et al., 2015; Zemel et al., 2013; Madras et al., 2018a; Zhao et al., 2019; Jang et al., 2024). Demographic parity, equal opportunity, equalized odds, and related misclassification-based criteria were then formalized as trainable or auditable constraints (Hardt et al., 2016; Zafar et al., 2017b;a; Agarwal et al., 2018; Celis et al., 2019), while calibration and multicalibration exposed tensions among statistical requirements that cannot generally be satisfied together (Kleinberg et al., 2016; Pleiss et al., 2017; Canetti et al., 2019; Hébert-Johnson et al., 2018; Hsu et al., 2022). Subgroup auditing further showed that coarse group parity can obscure disparities over structured subpopulations (Kearns et al., 2018; Speicher et al., 2018; Hashimoto et al., 2018).

A related algorithmic literature studies how to enforce, estimate, or certify these criteria under empirical and operational constraints, including reductions-based optimization, constrained empirical risk minimization, regression settings, and settings with limited or noisy protected-attribute information (Cotter et al., 2019; Agarwal et al., 2019; Mozannar et al., 2020; Celis et al., 2021; Mukherjee et al., 2020; Wu et al., 2022b;

Bendekgey & Sudderth, 2021; Taufiq et al., 2024). Surveys continue to organize much of the field around the comparison of such measurable criteria (Caton & Haas, 2024; Tang et al., 2023). At the same time, political-philosophical and legal analyses emphasize that statistical metrics already carry normative content: they encode particular views about equality of opportunity, discrimination, arbitrariness, and substantive fairness (Binns, 2018; Heidari et al., 2019; Arif Khan et al., 2022; Corbett-Davies et al., 2023; Long, 2021). In this paper, DP, EO, and Calibration are treated as meaningful surrogate constraints whose connection to harm allocation must be made explicit.

## 2.2 Welfare, Utility, and Preference-Aware Fairness

A complementary line of work evaluates algorithmic decisions through welfare, utility, preference, and burden rather than through observable-rate parity alone. These approaches ask how a policy changes the distribution of benefits, risks, and opportunities across individuals and groups (Heidari et al., 2018; Cousins, 2021; Hu & Chen, 2020; Kim et al., 2019; Milli et al., 2019; Williamson & Menon, 2019; Li & Liu, 2022; Liang & Lu, 2024; Pardeshi et al., 2024). This literature is directly connected to individualized error costs: once false positives and false negatives impose different burdens on different people, a statistical constraint also becomes an implicit cost-sensitive or decision-theoretic commitment (Menon & Williamson, 2018).

Welfare- and utility-based formulations also arise in online, sequential, and allocation settings, where fairness is expressed through welfare aggregation over rewards, regret, or feasible assignments (Hossain et al., 2021; Barman et al., 2023; Sawarni et al., 2023; Zhang et al., 2024; Siddique et al., 2020; Sim et al., 2021; Devic et al., 2023; Lobo et al., 2024). Similar issues appear in ranking, recommendation, and multi-sided platforms, where algorithmic systems allocate exposure, attention, and market opportunity among stakeholders with potentially conflicting objectives (Ge et al., 2022; Singh & Joachims, 2018; Wang et al., 2021; Do et al., 2021; Ge et al., 2021; Wu et al., 2022a; Greenwood et al., 2024; Chen et al., 2024). These works motivate a shift from metric satisfaction alone to the distribution of valued consequences.

## 2.3 Axiomatic Fairness and Fair Division

Fair division studies fairness as a property of allocations. Rather than beginning from prediction rates, it specifies axioms such as envy-freeness, proportionality, maximin-share guarantees, competitive-equilibrium-based criteria, and ordinal assignment principles, then analyzes their existence, approximation, and efficiency implications (Amanatidis et al., 2022; 2023; Aziz et al., 2022b; Lipton et al., 2004; Bouveret et al., 2005; Bouveret & Lemaître, 2016; Caragiannis et al., 2023; Budish, 2011; Bogomolnaia & Moulin, 2001; Segal-Halevi, 2020). This perspective provides comparison benchmarks for asking who receives what, relative to whom, and under which permissible complaint.

Algorithmic fair division has developed a large theory of approximate guarantees for indivisible goods, including EF1, EFX, maximin-share guarantees, Nash-social-welfare rules, and fairness under asymmetric entitlements or combinatorial constraints (Caragiannis et al., 2019b; Cole & Gkatzelis, 2015; Caragiannis et al., 2019a; Kurokawa et al., 2016; 2018; Garg & Taki, 2020; Farhadi et al., 2019; Aziz et al., 2022a; Barman & Biswas, 2018; Li & Vetta, 2021; Greco & Scarcello, 2020; Baklanov et al., 2021; Oh et al., 2021). Extensions to mixed divisible–indivisible resources, public decisions, public goods, constrained random assignment, chores, bads, and mixed resources are especially relevant for harm allocation, because classification errors and downstream decision burdens are allocative bads rather than goods (Bei et al., 2021b;a; Babichenko et al., 2024; Conitzer et al., 2017; Fain et al., 2018; Garg et al., 2021; Kawase et al., 2022; Bhaskar et al., 2020).

## 2.4 Fairness under Mediation, Uncertainty, and Bounded Rationality

A further line of work studies fairness when decisions are made under imperfect information, mediated through institutional pipelines, or shaped by downstream decision-makers whose knowledge and incentives differ from those of the model designer. Labels may be selectively observed, target variables may be imperfect proxies, sensitive attributes may be noisy, and final outcomes may depend on how models interact with human experts or organizational procedures (Kilbertus et al., 2020; Wei, 2021; Guerdan et al., 2023; Chouldechova

et al., 2018; Robertson et al., 2021; Madras et al., 2018b; Mozannar & Sontag, 2020; Charusaie et al., 2022; Mozannar et al., 2023; Ghoummaid & Shalit, 2024; Mehrotra & Vishnoi, 2022). Dataset construction itself is also part of the fairness problem, since measurements, labels, and categories are produced through institutional choices rather than simply observed from the world (Arif Khan et al., 2022).

Related work on delayed effects, performative prediction, reinforcement learning, and dynamic fairness constraints shows that fairness assessments can shift as decisions reshape populations, data distributions, and future observability (Liu et al., 2018; Zhang et al., 2019; Hardt & Mendler-Dünner, 2025; Brown et al., 2022; Deng et al., 2022; Rateike et al., 2024; Somerstep et al., 2024). Strategic classification and recourse add another layer of mediation: affected individuals may adapt to the rule, manipulate observable features, or face unequal abilities to comply with model-induced demands (Dong et al., 2018; Sundaram et al., 2023; Liu et al., 2022; Estornell et al., 2023; Von Kügelgen et al., 2022; Fokkema et al., 2024; Perello et al., 2025; Von Kügelgen et al., 2022). Bounded rationality in the present framework captures the corresponding implementation constraint for harm allocation: the mediator observes only noisy or coarse cost information and chooses from restricted policy classes, so the implementable harm allocation may differ from the axiomatic benchmark.

## 3 Preliminaries: Decision Rules and Error-Burden Primitives

We consider a binary decision problem over a finite evaluation population indexed by $i \in \{1, \ldots, n\}$. Each individual has features $X \in \mathcal{X}$, a group attribute $A \in \mathcal{A}$, and an evaluation label $Y \in \{0, 1\}$. An observed instance is written as $(x_i, a_i, y_i)$. Uppercase letters $(X, A, Y)$ denote random variables drawn from an underlying population distribution $\mathcal{D}$.

We use two levels of notation. Distribution-level quantities describe standard statistical fairness criteria such as group error rates. The fair-division analysis is applied to the finite evaluation population, where each individual contributes an error-event profile and, in Section 4, an individual cost vector. Empirical evaluations instantiate this finite-population object using the test sample.

**Scores and decision rules.** A scoring model produces a score $s : \mathcal{X} \to [0, 1]$, where larger values indicate a greater predicted likelihood that $Y = 1$. A decision rule, or policy, maps the information available at decision time to a binary action $\hat{Y} \in \{0, 1\}$, where $\hat{Y} = 1$ denotes a positive decision such as approval or acceptance, and $\hat{Y} = 0$ denotes a negative decision such as denial or rejection. To include randomized post-processing, we model the policy as a Markov kernel

$$\pi : \mathcal{X} \times \mathcal{A} \to [0, 1], \qquad \pi(x, a) = \Pr_\pi(\hat{Y} = 1 \mid X = x, A = a).$$

Deterministic rules are the special case in which $\pi(x, a) \in \{0, 1\}$ for all $(x, a)$. The group attribute may enter the policy because many group-fairness interventions, including group-specific thresholding and randomized post-processing, are defined at this stage even when the base score is group-blind.

**Error-event primitives.** The basic adverse events in the binary setting are false positives and false negatives:

$$\mathrm{FP} := \{\hat{Y} = 1, Y = 0\}, \qquad \mathrm{FN} := \{\hat{Y} = 0, Y = 1\}.$$

These events are not yet harms by themselves. They are error-event primitives. In Section 4, they are combined with individual-specific error costs to define expected harm.

For an observed instance $(x_i, a_i, y_i)$, the realized label $y_i$ is the evaluation target with respect to which false positives and false negatives are measured. Conditional on this observed instance, the policy randomization induces

$$e_i^{\mathrm{FP}}(\pi) := \Pr_\pi(\mathrm{FP} \mid x_i, a_i, y_i) = \mathbb{1}\{y_i = 0\}\pi(x_i, a_i),$$

$$e_i^{\mathrm{FN}}(\pi) := \Pr_\pi(\mathrm{FN} \mid x_i, a_i, y_i) = \mathbb{1}\{y_i = 1\}\big(1 - \pi(x_i, a_i)\big).$$

The pair

$$e_i(\pi) := \big(e_i^{\mathrm{FP}}(\pi), e_i^{\mathrm{FN}}(\pi)\big)$$

is the error-burden profile assigned to individual $i$ by policy $\pi$.

**Realized-label and expected-label accounting.** The main analysis uses the realized-label version above, which gives an ex post accounting of policy-induced error burdens on the evaluation population. This matches standard empirical evaluations of false positives and false negatives: $y_i$ is the label relative to which the deployed decision rule is assessed.

When the relevant target is uncertain, the same object can be written in an ex ante form. Let

$$\eta_i := \Pr_{\mathcal{D}}(Y = 1 \mid X = x_i, A = a_i).$$

Then the corresponding expected-label error primitives are

$$\bar{e}_i^{\mathrm{FP}}(\pi) = (1 - \eta_i)\pi(x_i, a_i), \qquad \bar{e}_i^{\mathrm{FN}}(\pi) = \eta_i\big(1 - \pi(x_i, a_i)\big).$$

Thus the expected-label formulation replaces the realized indicators $\mathbb{1}\{y_i = 0\}$ and $\mathbb{1}\{y_i = 1\}$ with conditional label probabilities $1 - \eta_i$ and $\eta_i$. The finite-population harm-allocation definitions below apply to either choice of error primitives.

**Population- and group-level rates.** For each group value $a \in \mathcal{A}$, the standard conditional error rates under policy $\pi$ are

$$\mathrm{FPR}_a(\pi) := \Pr_{\mathcal{D},\pi}(\hat{Y} = 1 \mid Y = 0, A = a), \qquad \mathrm{FNR}_a(\pi) := \Pr_{\mathcal{D},\pi}(\hat{Y} = 0 \mid Y = 1, A = a).$$

We also write

$$\mathrm{TPR}_a(\pi) = 1 - \mathrm{FNR}_a(\pi), \qquad \mathrm{TNR}_a(\pi) = 1 - \mathrm{FPR}_a(\pi).$$

Probabilities subscripted by $(\mathcal{D}, \pi)$ include randomness from both the population distribution and any randomization in the decision rule. In empirical evaluations, these rates are estimated by their sample analogues on the finite evaluation population.

**Scope of the allocation object.** This paper studies the allocation of error-induced burdens. A policy also allocates goods, such as positive decisions, and may generate broader downstream effects, such as delay, scrutiny, stigma, or opportunity costs. The formal analysis focuses on the component of these effects represented by false-positive and false-negative error events, together with the subjective costs attached to them. The resulting object is a finite vector of individual error burdens that becomes a harm allocation once individual costs are introduced.

## 4 Subjective Value: Individualized Error Costs and Harm Vectors

Error events can carry different burdens for different individuals. A false negative in a credit, employment, medical, or educational decision may impose financial loss, lost opportunity, delay, stigma, or safety risk, and the severity of these burdens may vary both across and within demographic groups. We represent this variation through individual-specific harm weights attached to the false-positive and false-negative error-event primitives defined in Section 3.

**Individualized error-cost weights.** Each individual $i$ is associated with a nonnegative cost vector

$$c_i := \big(c_i^{\mathrm{FP}}, c_i^{\mathrm{FN}}\big), \qquad c_i^{\mathrm{FP}} \geq 0, \quad c_i^{\mathrm{FN}} \geq 0.$$

The components $c_i^{\mathrm{FP}}$ and $c_i^{\mathrm{FN}}$ are harm weights for false-positive and false-negative error events. Their institutional source may vary by domain. They may be elicited from affected individuals, estimated from welfare losses, derived from legal compensation principles, calibrated through expert assessment, or specified through stakeholder-informed severity judgments. The framework treats these weights as normative inputs to harm accounting: once a cost representation is specified, the induced decision burdens can be compared across individuals and policies.

**Individual expected harm.** Fix a decision rule $\pi$. Let

$$e_i(\pi) = \big(e_i^{\mathrm{FP}}(\pi), e_i^{\mathrm{FN}}(\pi)\big)$$

denote the error-event profile assigned to individual $i$ under $\pi$. The expected harm borne by $i$ is the cost-weighted error burden

$$H_i(\pi) := c_i^{\mathrm{FP}} e_i^{\mathrm{FP}}(\pi) + c_i^{\mathrm{FN}} e_i^{\mathrm{FN}}(\pi) = \langle c_i, e_i(\pi) \rangle. \tag{4.1}$$

This linear form gives a tractable first-order representation of error-induced burden. It separates the probability of an adverse decision event from the weight assigned to that event, allowing the same policy-induced error profile to have different harm implications for different individuals.

**Harm-allocation vector.** The harm allocation induced by $\pi$ on the finite evaluation population is

$$H(\pi) := \big(H_1(\pi), \ldots, H_n(\pi)\big) \in \mathbb{R}_+^n. \tag{4.2}$$

Fairness axioms in the next section are applied to this vector and to the underlying evaluated error-event profiles. The latter are important because fair-division comparisons require asking how one individual's cost vector evaluates another individual's assigned error burden. For this purpose, we write

$$H_{i \leftarrow j}(\pi) := c_i^{\mathrm{FP}} e_j^{\mathrm{FP}}(\pi) + c_i^{\mathrm{FN}} e_j^{\mathrm{FN}}(\pi) = \langle c_i, e_j(\pi) \rangle. \tag{4.3}$$

Thus $H_i(\pi) = H_{i \leftarrow i}(\pi)$, while $H_{i \leftarrow j}(\pi)$ measures how individual $i$ would evaluate the error-event burden assigned to individual $j$ using $i$'s own cost vector.

**Cost heterogeneity and statistical aggregation.** Cost heterogeneity can arise between demographic groups or among individuals within the same group. Because DP, EO, and Calibration constrain group-level rates or score behavior, they operate on aggregated statistical objects. Harm allocation operates on the cost-weighted individual burdens induced by those objects. The relationship between the two depends on the structure of the cost vectors: when costs are homogeneous or aligned with the relevant statistical partitions, rate-based constraints can track harm-based comparisons more closely; when costs vary within or across those partitions, the same rates can induce different harm allocations.

**Aggregate harm summaries.** For efficiency and welfare comparisons, we use scalar summaries of the harm vector, including mean harm and tail harm:

$$\overline{H}(\pi) := \frac{1}{n} \sum_{i=1}^{n} H_i(\pi), \qquad H^{(q)}(\pi) := \mathrm{Quantile}_q\big(\{H_i(\pi)\}_{i=1}^n\big).$$

Here $H^{(q)}(\pi)$ denotes the $q$-quantile of individual harms, such as $q = 0.9$ for upper-tail burden. These summaries provide efficiency and concentration diagnostics alongside the fair-division axioms defined below.

## 5 Axiomatic Fair Division of Harm

We evaluate policy-induced harm allocations using two canonical fair-division diagnostics for allocative bads: envy-freeness and proportionality. Envy-freeness captures comparative grievance: whether an individual, using their own error-cost weights, evaluates another person's assigned error-event burden as less costly than their own. Proportionality captures share-based protection: whether an individual's own burden exceeds an equal share of the total burden, measured in that individual's cost units. Together, these diagnostics separate relational unfairness from burden concentration.

Throughout this section, $H_{i \leftarrow j}(\pi)$ denotes the cross-evaluated harm defined in Equation equation 4.3. Thus $H_{i \leftarrow i}(\pi) = H_i(\pi)$ is individual $i$'s own harm, while $H_{i \leftarrow j}(\pi)$ evaluates individual $j$'s assigned error-event profile using individual $i$'s cost vector.

**Envy-freeness.** For allocative bads, individual $i$ has an envy complaint against individual $j$ when $i$'s own assigned burden is larger than the burden that $i$ assigns to $j$'s error-event profile.

**Definition 5.1** (Envy-freeness). A policy $\pi$ is *envy-free* (EF) if, for all individuals $i, j$,

$$H_{i \leftarrow i}(\pi) \leq H_{i \leftarrow j}(\pi).$$

The corresponding pairwise violation is

$$\text{EnvyGap}_{i,j}(\pi) := \left[ H_{i \leftarrow i}(\pi) - H_{i \leftarrow j}(\pi) \right]_+.$$

We use the aggregate scores

$$\text{Envy}^{\max}(\pi) := \max_{i,j} \text{EnvyGap}_{i,j}(\pi), \qquad \text{Envy}^{\text{avg}}(\pi) := \frac{1}{n(n-1)} \sum_{i \neq j} \text{EnvyGap}_{i,j}(\pi),$$

and

$$\text{EnvyFrac}(\pi) := \frac{1}{n(n-1)} \sum_{i \neq j} \mathbb{1}\{\text{EnvyGap}_{i,j}(\pi) > 0\}.$$

**Definition 5.2** ($\delta$-envy-freeness). For $\delta \geq 0$, a policy $\pi$ is $\delta$-*envy-free* if

$$\text{Envy}^{\max}(\pi) \leq \delta.$$

**Proportionality.** Proportionality gives a share-based upper-bound diagnostic for harms. Individual $i$ evaluates the entire policy-induced error burden as

$$\sum_{j=1}^{n} H_{i \leftarrow j}(\pi),$$

so the proportional benchmark for $i$ is the equal share of that total in $i$'s own cost units.

**Definition 5.3** (Proportionality). A policy $\pi$ is *proportional* (PROP) if, for all individuals $i$,

$$H_{i \leftarrow i}(\pi) \leq \frac{1}{n} \sum_{j=1}^{n} H_{i \leftarrow j}(\pi).$$

The individualized proportionality excess is

$$\text{PropExcess}_i(\pi) := \left[ H_{i \leftarrow i}(\pi) - \frac{1}{n} \sum_{j=1}^{n} H_{i \leftarrow j}(\pi) \right]_+.$$

We use

$$\text{Prop}^{\max}(\pi) := \max_i \text{PropExcess}_i(\pi), \qquad \text{Prop}^{\text{avg}}(\pi) := \frac{1}{n} \sum_{i=1}^{n} \text{PropExcess}_i(\pi).$$

**Definition 5.4** ($\delta$-proportionality). For $\delta \geq 0$, a policy $\pi$ is $\delta$-*proportional* if

$$\text{Prop}^{\max}(\pi) \leq \delta.$$

**Empirical cost-stratum envy.** Exact pairwise envy requires $O(n^2)$ cross-evaluations. In large empirical evaluations, we use a cost-stratum version that preserves the same comparative structure at a coarser resolution. Let $G(i) \in \{1, \ldots, K\}$ denote the cost-stratum index obtained from quantiles of $\log(c_i^{\text{FP}}/c_i^{\text{FN}})$, and let

$$S_g := \{i : G(i) = g\}, \qquad \bar{c}_g := (\bar{c}_g^{\text{FP}}, \bar{c}_g^{\text{FN}})$$

be the corresponding stratum and mean cost vector. On an evaluation split, define

$$\widehat{\Pr}(\text{FP} \mid g) := \frac{1}{|S_g|} \sum_{i \in S_g} \mathbb{1}\{y_i = 0, \hat{y}_i = 1\}, \qquad \widehat{\Pr}(\text{FN} \mid g) := \frac{1}{|S_g|} \sum_{i \in S_g} \mathbb{1}\{y_i = 1, \hat{y}_i = 0\}.$$

The cross-evaluated stratum harm is

$$\widetilde{H}_{g \leftarrow g'}(\pi) := \bar{c}_g^{\text{FP}} \widehat{\Pr}(\text{FP} \mid g') + \bar{c}_g^{\text{FN}} \widehat{\Pr}(\text{FN} \mid g').$$

The stratum-level envy gap is

$$\widetilde{\text{EnvyGap}}_{g,g'}(\pi) := \left[ \widetilde{H}_{g \leftarrow g}(\pi) - \widetilde{H}_{g \leftarrow g'}(\pi) \right]_+,$$

with aggregate score

$$\widetilde{\text{Envy}}^{\max}(\pi) := \max_{g,g'} \widetilde{\text{EnvyGap}}_{g,g'}(\pi).$$

This score measures comparative burden across cost strata. The robustness analysis in Appendix E compares this scalable proxy with individual-pair envy on smaller evaluation subsets.

**Efficiency loss.** When comparing a fairness-constrained policy with a reference policy, we report the mean-harm increase

$$\text{PoF}(\pi^{\text{fair}}; \pi^\star) := \overline{H}(\pi^{\text{fair}}) - \overline{H}(\pi^\star),$$

where $\pi^\star$ is a reference policy within the chosen policy class and $\pi^{\text{fair}}$ is the policy selected under the target fairness constraint or violation score. We also report the ratio

$$\frac{\overline{H}(\pi^{\text{fair}})}{\overline{H}(\pi^\star)}$$

when a scale-free comparison is more informative.

## 6 Institutional Bounded Rationality and Intermediation Constraints

Harm-allocation fairness is implemented through institutions. A platform, agency, employer, lender, or regulator selects a decision rule under limited information about individual harm weights and under constraints on the policies it can deploy. We call this setting *institutional bounded rationality*. In this paper, the relevant bounds are informational and implementational: the intermediary observes an imperfect representation of error costs and chooses from a restricted policy class.

**Intermediary choice problem.** Let $\Pi$ denote a feasible class of decision rules. The intermediary selects $\pi \in \Pi$ using the available data, the score function, the permitted use of group attributes, and an observed cost representation. For a specified harm-weight vector

$$c = \{c_i\}_{i=1}^n, \qquad c_i = (c_i^{\text{FP}}, c_i^{\text{FN}}),$$

the policy induces the harm allocation $H(\pi; c)$ and the axiom-violation scores defined in Section 5. We write

$$\text{Viol}(\pi; c)$$

for a generic violation functional, such as $\text{Envy}^{\max}(\pi)$ or $\text{Prop}^{\max}(\pi)$, with the dependence on $c$ made explicit.

**Information bounds on harm weights.** The intermediary typically works with a proxy or institutional representation

$$\hat{c} = \{\hat{c}_i\}_{i=1}^n$$

of the adopted harm weights. This proxy may come from elicitation, administrative data, legal or policy categories, expert severity weights, stakeholder-informed assessments, or a combination of these sources. To represent the remaining ambiguity, define the compatible cost set

$$\mathcal{C}(\hat{c}, \epsilon) := \left\{ c = \{c_i\}_{i=1}^n : \|c_i - \hat{c}_i\| \le \epsilon \text{ for all } i \right\}, \tag{6.1}$$

where the norm and radius $\epsilon \ge 0$ encode the resolution of the available cost information. Additive error in cost space and multiplicative error in log-cost space are both instances of this representation.

A coarser information structure arises when the intermediary observes only a cost stratum

$$g(i) \in \{1, \dots, K\}$$

and assigns a stratum-level proxy $\hat{c}_{g(i)}$ to individual $i$. Then the available representation is

$$\hat{c}_i = \hat{c}_{g(i)}, \qquad c_i \in \mathcal{C}_{g(i)}.$$

This captures settings where institutions operate through categories, score bands, administrative classes, or survey-derived profiles. Within-stratum cost variation becomes part of the information bound faced by the intermediary.

**Policy-class bounds.** The intermediary also faces restrictions on the set of deployable decision rules. We write the implementable class as

$$\Pi_{\mathrm{bd}} \subseteq \Pi.$$

Examples include single-threshold rules,

$$\hat{Y} = \mathbb{1}\{s(X) \ge \tau\},$$

group-specific thresholds,

$$\hat{Y} = \mathbb{1}\{s(X) \ge \tau_A\},$$

and low-dimensional randomized post-processing rules $\pi(s, a)$. These restrictions reflect auditing requirements, legal constraints, operational simplicity, and the need to deploy stable rules that can be communicated and monitored.

**Achievable violation under bounded rationality.** Given an observed proxy $\hat{c}$, an ambiguity radius $\epsilon$, and an implementable policy class $\Pi_{\mathrm{bd}}$, define the bounded-rationality benchmark

$$\mathrm{Ach}(\hat{c}, \epsilon; \Pi_{\mathrm{bd}}) := \inf_{\pi \in \Pi_{\mathrm{bd}}} \sup_{c \in \mathcal{C}(\hat{c}, \epsilon)} \mathrm{Viol}(\pi; c). \tag{6.2}$$

This quantity measures the smallest violation level the intermediary can guarantee across the cost representations compatible with its information. The benchmark connects exact axiomatic fairness to implementable fairness: even when EF or PROP defines the target, the attainable violation depends on the precision of the cost representation and the expressive power of the policy class.

The same notation separates two sources of approximation. The information component is captured by the size and structure of $\mathcal{C}(\hat{c}, \epsilon)$: noisier, coarser, or more aggregated cost information expands the set of compatible harm-weight representations. The implementation component is captured by $\Pi_{\mathrm{bd}}$: simpler policy classes restrict the allocations that can be induced from a fixed score and dataset.

**Approximation floors.** Bounded rationality turns harm-based fairness into an approximation problem. For a violation functional Viol, an information model $\mathcal{C}(\hat{c}, \epsilon)$, and a bounded policy class $\Pi_{\mathrm{bd}}$, lower bounds of the form

$$\mathrm{Ach}(\hat{c}, \epsilon; \Pi_{\mathrm{bd}}) \ge f(\epsilon, \Pi_{\mathrm{bd}})$$

describe an irreducible violation floor induced by imperfect cost information and restricted implementation. These floors formalize the gap between an axiomatic harm-allocation target and the fairness level that an intermediary can guarantee under institutional constraints.

# 7 Statistical Fairness Metrics as Surrogate Conditions

Statistical fairness metrics constrain observable rates, scores, or conditional distributions. Harm-allocation axioms evaluate cost-weighted individual error burdens. The connection between the two depends on how individual harm weights align with the statistical partitions used by the metric. This section formalizes that connection: first through positive alignment conditions, then through counterexamples showing how the surrogate relation breaks under cost heterogeneity, and finally through an approximation-floor result under institutional bounded rationality.

**Statistical metrics.** Let $A \in \mathcal{A}$ denote a group attribute, let $S = s(X)$ be a score, and let $\pi$ be a possibly randomized decision rule. We use the following standard criteria:

$$\text{DP}: \quad \Pr_{\mathcal{D},\pi}(\hat{Y} = 1 \mid A = a) \text{ is constant in } a,$$

$$\text{EO}: \quad \Pr_{\mathcal{D},\pi}(\hat{Y} = 1 \mid Y = y, A = a) \text{ is constant in } a \text{ for each } y \in \{0, 1\},$$

with equal opportunity as the one-sided version requiring equality of $\text{TPR}_a(\pi)$. For score-based calibration, we use groupwise calibration of $S$:

$$\Pr_{\mathcal{D}}(Y = 1 \mid S = u, A = a) = u$$

for all relevant score values $u$ and groups $a$, with the usual binned analogue in empirical settings.

## 7.1 Surrogate alignment under cost structure

The basic bridge between statistical rates and harm allocation is obtained when error-cost weights are structured by group. Suppose that, within group $a$, the harm weights are constant:

$$c_i^{\text{FP}} = c_a^{\text{FP}}, \qquad c_i^{\text{FN}} = c_a^{\text{FN}} \quad \text{whenever } A_i = a.$$

Then the group-average harm is

$$\overline{H}_a(\pi) := \mathbb{E}_{\mathcal{D},\pi}[H_i(\pi) \mid A = a] = c_a^{\text{FP}} \Pr(Y = 0 \mid A = a)\text{FPR}_a(\pi) + c_a^{\text{FN}} \Pr(Y = 1 \mid A = a)\text{FNR}_a(\pi). \quad (7.1)$$

This identity identifies the structural conditions under which rate constraints become informative about harm. EO constrains the two error-rate terms in equation 7.1; EOp constrains the false-negative component; calibration aligns score values with expected-label accounting inside score bins. When the coefficients multiplying these rates are comparable across the relevant groups or bins, statistical constraints can act as low-dimensional surrogates for group-average harm comparisons.

**Proposition 7.1** (Surrogate alignment). *Assume group-homogeneous harm weights. If two groups $a$ and $b$ satisfy*

$$c_a^{\text{FP}} \Pr(Y = 0 \mid A = a) = c_b^{\text{FP}} \Pr(Y = 0 \mid A = b), \qquad c_a^{\text{FN}} \Pr(Y = 1 \mid A = a) = c_b^{\text{FN}} \Pr(Y = 1 \mid A = b),$$

*then EO equality between $a$ and $b$ implies equality of group-average harm:*

$$\overline{H}_a(\pi) = \overline{H}_b(\pi).$$

*If the second equality holds and EOp holds, then the false-negative component of group-average harm is equalized across the two groups.*

*Proof.* Substitute the equalized error rates into equation 7.1. The EOp case follows by retaining only the false-negative term. □

The proposition gives a positive boundary for surrogate use. Statistical criteria become harm-informative when the relevant harm weights, base rates, and partitions align. The negative results below identify what breaks when that alignment is absent.

**Theorem 7.2** (Need for cost-structure restrictions)**.** *Let $M$ be any statistical constraint whose value depends on the distribution of observable quantities such as $(A, Y, S, \hat{Y})$ and not on the harm-weight vector $c$. Suppose an instance satisfying $M$ contains two individuals $i$ and $j$ with different error-event profiles,*

$$e_i(\pi) \neq e_j(\pi).$$

*Then, for every $B > 0$, there exists a nonnegative harm-weight vector $c$ such that $M$ remains unchanged and*

$$\text{Envy}^{\max}(\pi; c) \geq B.$$

*Proof.* Since $M$ is independent of $c$, changing harm weights leaves the statistical constraint unchanged. If $e_i^{\text{FN}}(\pi) > e_j^{\text{FN}}(\pi)$, set $c_i^{\text{FN}} = C$ and $c_i^{\text{FP}} = 0$. Then

$$H_{i \leftarrow i}(\pi) - H_{i \leftarrow j}(\pi) = C\big(e_i^{\text{FN}}(\pi) - e_j^{\text{FN}}(\pi)\big),$$

which exceeds $B$ for sufficiently large $C$. The false-positive case is identical with the two components exchanged. □

## 7.2 Counterexamples: metric satisfaction and harm-axiom violation

The preceding theorem uses a generic cost-scaling argument. The next result gives explicit instances for the standard fairness metrics.

**Theorem 7.3** (Counterexamples under heterogeneous costs)**.** *For each metric family*

$$M \in \{\text{DP}, \text{EO}, \text{EOp}, \text{Calibration}\}$$

*and for every tolerance $\eta \geq 0$ and bound $B > 0$, there exist a distribution $\mathcal{D}$, a policy $\pi$, and a nonnegative harm-weight vector $c$ such that*

$$M(\pi) \leq \eta, \qquad \text{Envy}^{\max}(\pi; c) \geq B, \qquad \text{Prop}^{\max}(\pi; c) \geq B.$$

*For calibration, $M$ denotes the groupwise calibration error of the score $S$, while the policy $\pi$ maps calibrated scores to decisions.*

*Proof.* For DP, EO, and EOp, use two groups with equal base rates and a randomized post-processing rule whose group-conditional true-positive and false-positive rates are matched across groups. Select two same-group individuals with different false-negative probabilities and scale the false-negative harm weight of the higher-burden individual. The statistical metric remains fixed because the costs do not enter the rate constraint, while the envy gap and proportionality excess grow with the scaling factor.

For calibration, use a calibrated score $S \in \{u_0, u_1\}$ with $\Pr(Y = 1 \mid S = u, A = a) = u$ and a randomized decision rule $\Pr_\pi(\hat{Y} = 1 \mid S = u) = u$. Two positively labeled individuals with scores $u_0 < u_1$ have different false-negative probabilities. Scaling the false-negative harm weight of the lower-score individual yields the same divergence in envy and proportionality violation. The full finite constructions are given in Appendix A. □

## 7.3 Incompatibility as an allocation tradeoff

Classical incompatibility results show that calibration and EO impose conflicting statistical requirements under unequal base rates. In the harm-allocation framework, this conflict matters because relaxing either requirement changes the false-positive and false-negative terms that enter Equation equation 7.1.

**Proposition 7.4** (Calibration–EO incompatibility and harm allocation)**.** *Let $A \in \{a, b\}$ with unequal base rates*

$$\mu_a := \Pr(Y = 1 \mid A = a) \neq \mu_b := \Pr(Y = 1 \mid A = b).$$

*For a nondegenerate classifier, groupwise calibration of the score and EO for the induced classifier cannot both hold. Any relaxation that changes $\text{FPR}_a, \text{FPR}_b$ or $\text{FNR}_a, \text{FNR}_b$ changes the corresponding group-average harm terms in equation 7.1. Under heterogeneous harm weights, these rate movements can increase envy or proportionality violation even when the relaxed statistical metric improves.*

*Proof.* Under EO, write

$$\text{TPR}_a = \text{TPR}_b =: \text{TPR}, \qquad \text{FPR}_a = \text{FPR}_b =: \text{FPR}.$$

Bayes' rule gives

$$\Pr(Y = 1 \mid \hat{Y} = 1, A = g) = \frac{\mu_g \text{TPR}}{\mu_g \text{TPR} + (1 - \mu_g)\text{FPR}}, \qquad g \in \{a, b\}.$$

When $\mu_a \neq \mu_b$, the two positive predictive values differ for every nondegenerate classifier. Groupwise calibration requires these predictive values to agree at the same score level, giving the standard incompatibility. The harm-allocation implication follows by applying equation 7.1: changes in the error-rate components alter cost-weighted group harm, and the cross-evaluated quantities $H_{i \leftarrow j}(\pi)$ translate those changes into EF and PROP violations for suitable cost profiles. Detailed constructions are in Appendix A. □

### 7.4 Lower bounds under bounded rationality

The previous results concern surrogate validity for a fixed harm-weight representation. Institutional bounded rationality adds a second source of approximation: the intermediary observes only an imperfect cost representation and chooses within a restricted policy class.

**Theorem 7.5** (Minimal approximation floor). *Fix $\epsilon \in (0, 1)$. Consider two individuals with labels $y_1 = 1$ and $y_2 = 0$. The intermediary observes the proxy costs*

$$\hat{c}_1 = \hat{c}_2 = (1, 1)$$

*and the compatible cost set*

$$\mathcal{C}(\hat{c}, \epsilon) = \left\{ c : c_i^{\text{FP}}, c_i^{\text{FN}} \in [1 - \epsilon, 1 + \epsilon] \text{ for } i = 1, 2 \right\}.$$

*Let the bounded policy class enforce identical treatment:*

$$\Pi_{\text{bd}} = \left\{ \pi : \Pr_\pi(\hat{Y} = 1 \mid 1) = \Pr_\pi(\hat{Y} = 1 \mid 2) =: \alpha, \ \alpha \in [0, 1] \right\}.$$

*For $\text{Viol}(\pi; c) = \text{Envy}^{\max}(\pi; c)$,*

$$\text{Ach}(\hat{c}, \epsilon; \Pi_{\text{bd}}) = \inf_{\pi \in \Pi_{\text{bd}}} \sup_{c \in \mathcal{C}(\hat{c}, \epsilon)} \text{Envy}^{\max}(\pi; c) = \epsilon.$$

*Proof.* Under identical treatment,

$$e_1^{\text{FN}}(\pi) = 1 - \alpha, \qquad e_2^{\text{FP}}(\pi) = \alpha.$$

The two possible envy gaps are

$$\text{EnvyGap}_{1,2}(\pi; c) = \left[ c_1^{\text{FN}}(1 - \alpha) - c_1^{\text{FP}}\alpha \right]_+,$$

$$\text{EnvyGap}_{2,1}(\pi; c) = \left[ c_2^{\text{FP}}\alpha - c_2^{\text{FN}}(1 - \alpha) \right]_+.$$

Maximizing each linear expression over the uncertainty set gives

$$\sup_{c \in \mathcal{C}(\hat{c}, \epsilon)} \text{EnvyGap}_{1,2}(\pi; c) = [1 + \epsilon - 2\alpha]_+,$$

and

$$\sup_{c \in \mathcal{C}(\hat{c}, \epsilon)} \text{EnvyGap}_{2,1}(\pi; c) = [2\alpha - (1 - \epsilon)]_+.$$

Therefore

$$\sup_{c \in \mathcal{C}(\hat{c}, \epsilon)} \text{Envy}^{\max}(\pi; c) = \max\{[1 + \epsilon - 2\alpha]_+, [2\alpha - (1 - \epsilon)]_+\}.$$

The minimum over $\alpha \in [0, 1]$ is attained at $\alpha = 1/2$, where both terms equal $\epsilon$. Hence

$$\text{Ach}(\hat{c}, \epsilon; \Pi_{\text{bd}}) = \epsilon.$$

□

# 8 Experiments

## 8.1 Common Experimental Design and Evaluation Metrics

The experiments evaluate when statistical fairness criteria act as reliable surrogates for harm-allocation fairness under heterogeneous error costs, imperfect cost information, and different intervention families. All experiments use ACS 2018 tasks from Folktables, restricted to California for memory control. We consider three binary prediction tasks, abbreviated in tables as INC (ACSINCOME), COV (ACSPUBLICCOVERAGE), and EMP (ACSEMPLOYMENT). The sensitive attribute is SEX; it is used as the group variable for auditing or group-aware post-processing and is removed from the feature set. For each task and random seed, we subsample before conversion to numpy and use stratified train/validation/test splits. Experiments 1–3 use logistic regression (LR) and histogram gradient boosting (HGB) score models, while Experiment 4 uses a shared linear logistic classifier to compare fairness interventions under a common base learner.

All harm-based evaluations use realized-label false-positive and false-negative burdens. For a policy $\pi$ producing predictions $\hat{y}$, individual harm is

$$H_i(\pi) = c_i^{\text{FP}} \mathbb{1}\{\text{FP}_i\} + c_i^{\text{FN}} \mathbb{1}\{\text{FN}_i\}.$$

The default heterogeneous cost regime is generated by

$$m_i = \begin{cases} e^{\gamma} & (A_i = 1), \\ e^{-\gamma} & (A_i = 0), \end{cases} \qquad u_i \sim \mathcal{N}(0, \sigma^2), \qquad c_i^{\text{FP}} = m_i e^{u_i}, \qquad c_i^{\text{FN}} = m_i e^{-u_i}.$$

This controlled synthetic cost family separates between-group harm scale, governed by $\gamma$, from within-group FP–FN trade-off heterogeneity, governed by $\sigma$. Unless otherwise stated, the main heterogeneous regime is $(\gamma, \sigma) = (2, 1)$, with a homogeneous-cost sanity check at $(\gamma, \sigma) = (0, 0)$.

We report both statistical surrogate metrics and harm-allocation diagnostics. The surrogate metrics are accuracy, DP gap, EO gap, and expected calibration error (ECE). The harm diagnostics are mean harm, maximum harm, tail excess, and cost-stratum envy. Cost-stratum envy is computed by partitioning individuals into quantiles of

$$\log(c_i^{\text{FP}}/c_i^{\text{FN}})$$

and evaluating the maximum cross-stratum envy score $\widetilde{\text{Envy}}^{\max}$ from Section 5. Unless otherwise stated, we use $K = 80$ cost strata. Additional robustness analyses in Appendix E vary the cost regime and compare the cost-stratum proxy with individual-level envy on smaller evaluation subsets.

## 8.2 Experiment 1: Surrogate–Axiom Alignment under Cost Heterogeneity

**Experimental setup.** Experiment 1 tests whether improving a statistical surrogate also improves harm-allocation fairness when error costs vary across individuals. From each trained score model, we construct four policy families: **BASE**, using a global threshold at 0.5; **DP**, using group-specific thresholds chosen through target acceptance rates; **EO**, using group-specific thresholds selected to reduce EO gap; and **CAL**, using isotonic calibration with a mixing parameter before thresholding. Candidate policies are selected on validation and evaluated on test. We report a homogeneous-cost sanity check, a strong-heterogeneity regime, and a within-family discordance rate measuring how often a better surrogate score coincides with worse cost-stratum envy.

**Results and analysis.** Table 1 shows that the homogeneous-cost setting produces near-zero cost-stratum envy because the cost-stratum comparison has no meaningful FP–FN preference variation to separate. Under heterogeneous costs, the relationship between statistical surrogates and harm-based envy becomes criterion-dependent. DP-selected policies tend to reduce cost-stratum envy relative to the baseline in this setup, EO-selected policies often move envy upward, and calibration post-processing produces smaller and mixed changes. The discordance panel shows that even within a fixed policy family, improving the statistical surrogate can move cost-stratum envy in the opposite direction.

| Task | Model | $(\gamma=0, \sigma=0)$: envy | | | |
|---|---|---|---|---|---|
| | | BASE | DP | EO | CAL |
| EMP | HGB | 0.0000 | 0.0000 | 0.0000 | 0.0000 |
| | LR | 0.0000 | 0.0000 | 0.0000 | 0.0000 |
| INC | HGB | 0.0000 | 0.0000 | 0.0000 | 0.0000 |
| | LR | 0.0000 | 0.0000 | 0.0000 | 0.0000 |
| COV | HGB | 0.0000 | 0.0000 | 0.0000 | 0.0000 |
| | LR | 0.0000 | 0.0000 | 0.0000 | 0.0000 |

(a)

| Task | Model | $(\gamma=2, \sigma=1)$: envy | | | |
|---|---|---|---|---|---|
| | | BASE | DP | EO | CAL |
| EMP | HGB | 6.2972 | 3.8361 | 8.2062 | 6.2963 |
| | LR | 6.4179 | 3.8513 | 7.4175 | 6.4179 |
| INC | HGB | 6.1190 | 4.6799 | 7.1064 | 6.1164 |
| | LR | 6.7336 | 5.1740 | 9.4393 | 6.5494 |
| COV | HGB | 6.8917 | 5.6462 | 8.1721 | 6.6435 |
| | LR | 8.6135 | 5.7380 | 8.4219 | 8.0447 |

(b)

| Task | Model | $(\gamma=2, \sigma=1)$: $\Delta$ (BASE) | | |
|---|---|---|---|---|
| | | $\Delta$DP | $\Delta$EO | $\Delta$CAL |
| EMP | HGB | -2.4611 | 1.9090 | -0.0009 |
| | LR | -2.5666 | 0.9996 | 0.0000 |
| INC | HGB | -1.4391 | 0.9874 | -0.0026 |
| | LR | -1.5596 | 2.7057 | -0.1842 |
| COV | HGB | -1.2455 | 1.2804 | -0.2482 |
| | LR | -2.8755 | -0.1916 | -0.5688 |

(c)

| Task | Model | $(\gamma=2, \sigma=1)$: discordance rate | | |
|---|---|---|---|---|
| | | DP | EO | CAL |
| EMP | HGB | 0.4921 | 0.8889 | 0.4000 |
| | LR | 0.5556 | 0.6667 | 0.2333 |
| INC | HGB | 0.4921 | 0.7778 | 0.3333 |
| | LR | 0.4603 | 0.6667 | 0.1000 |
| COV | HGB | 0.3492 | 0.2222 | 0.5000 |
| | LR | 0.3333 | 0.6667 | 0.4000 |

(d)

Table 1: Summary of statistical-fairness metrics and harm-based fairness outcomes across tasks and models, averaged over three random seeds, under homogeneous and heterogeneous subjective-cost regimes. Envy is reported as max cost-stratum envy $\widetilde{\text{Envy}}^{\max}$ using $K = 80$.

## 8.3 Experiment 2: Bounded Rationality under Information and Policy Constraints

**Experimental setup.** Experiment 2 operationalizes institutional bounded rationality. Scores and true test costs are fixed under the heterogeneous regime $(\gamma, \sigma) = (2, 1)$, but the intermediary selects policies on validation using imperfect cost information. We vary multiplicative cost-proxy noise $\epsilon \in \{0, 0.25, 0.5, 1.0\}$ and coarse cost information through proxy strata $k \in \{0, 2, 5, 10\}$, where $k = 0$ denotes individual-level proxy costs. The intermediary chooses from either global thresholds or group-specific thresholds and minimizes proxy cost-stratum envy on validation. We report realized test envy under true costs for four representative information regimes: oracle, noisy, coarse, and noisy+coarse.

**Results and analysis.** Tables 2–3 show persistent residual envy even when the intermediary observes oracle cost information, reflecting the restriction imposed by simple threshold policy classes. Noisy and coarse proxies can change the selected threshold in ways that increase or decrease realized true-cost envy, and group-specific thresholds do not uniformly improve the harm-based objective. The empirical pattern matches the bounded-rationality formulation: harm-based targeting remains constrained by both the resolution of cost information and the expressive capacity of the implementable policy class.

## 8.4 Experiment 3: Axiomatic Profiles of DP, EO, and Calibration

**Experimental setup.** Experiment 3 compares the harm profiles induced by targeting different statistical notions within the same post-processing template. For each task, model, and seed, we report representative **BASE**, **CAL**, **DP**, and **EO** policies. DP, EO, and CAL candidates are selected on validation by imposing a small feasibility tolerance on the corresponding surrogate metric and then choosing the most accurate feasible candidate, with best-effort selection when the tolerance is infeasible. Harm metrics are evaluated on the test set under fixed heterogeneous costs.

**Results and analysis.** Tables 4–5 show non-monotone profiles across statistical and harm-based criteria. Policies that improve their targeted surrogate do not necessarily improve cost-stratum envy, mean harm,

| Task | Model | $\delta$ | | | | Task | Model | $\Delta$ | | | |
|------|-------|--------|-------|--------|--------------|------|-------|--------|---------|--------|--------------|
| | | Oracle | Noisy | Coarse | Noisy+coarse | | | Oracle | Noisy | Coarse | Noisy+coarse |
| EMP | HGB | 4.3838 | 6.4273 | 5.9476 | 6.2358 | EMP | HGB | -0.8429 | 1.2007 | 0.7210 | 1.0092 |
| | LR | 5.0616 | 5.6175 | 8.5032 | 5.7925 | | LR | -0.3962 | 0.1596 | 3.0454 | 0.3347 |
| INC | HGB | 6.9679 | 6.3603 | 7.0954 | 7.7142 | INC | HGB | 0.9348 | 0.3271 | 1.0623 | 1.6810 |
| | LR | 8.0557 | 7.1278 | 7.6640 | 8.3580 | | LR | 1.7349 | 0.8070 | 1.3432 | 2.0372 |
| COV | HGB | 13.7500 | 11.4117 | 14.4939 | 15.8237 | COV | HGB | 1.8620 | -0.4762 | 2.6059 | 3.9357 |
| | LR | 13.2415 | 13.8218 | 14.8084 | 15.4885 | | LR | -1.4834 | -0.9031 | 0.0836 | 0.7636 |

|   (a)   |   (b)   |
|---------|---------|

Table 2: Realized envy-gap $\delta$ and change $\Delta = \delta - \delta_{\mathbf{BASE}}$ under true costs $(\gamma, \sigma) = (2, 1)$ for single global-threshold policies, averaged over three random seeds. Envy is computed as the maximum cost-stratum envy using a fine-grained partition into 80 cost strata.

| Task | Model | $\delta$ | | | | Task | Model | $\Delta$ | | | |
|------|-------|---------|---------|---------|--------------|------|-------|--------|---------|--------|--------------|
| | | Oracle | Noisy | Coarse | Noisy+coarse | | | Oracle | Noisy | Coarse | Noisy+coarse |
| EMP | HGB | 7.2337 | 6.5505 | 7.3212 | 4.9492 | EMP | HGB | 2.0070 | 1.3238 | 2.0946 | -0.2774 |
| | LR | 5.8432 | 5.8149 | 8.7682 | 6.3713 | | LR | 0.3854 | 0.3571 | 3.3104 | 0.9135 |
| INC | HGB | 7.4117 | 5.5227 | 6.7408 | 7.5837 | INC | HGB | 1.3786 | -0.5105 | 0.7077 | 1.5506 |
| | LR | 6.6658 | 5.5445 | 7.4765 | 8.1880 | | LR | 0.3451 | -0.7763 | 1.1557 | 1.8672 |
| COV | HGB | 19.2726 | 16.7099 | 18.5463 | 18.5463 | COV | HGB | 7.3846 | 4.8220 | 6.6584 | 6.6584 |
| | LR | 16.5319 | 12.6476 | 15.4844 | 15.5148 | | LR | 1.8070 | -2.0773 | 0.7595 | 0.7900 |

|   (a)   |   (b)   |
|---------|---------|

Table 3: Realized envy-gap $\delta$ and change $\Delta = \delta - \delta_{\mathbf{BASE}}$ under true costs $(\gamma, \sigma) = (2, 1)$ for group-specific threshold policies, averaged over three random seeds. Envy is computed as the maximum cost-stratum envy using a fine-grained partition into 80 cost strata.

maximum harm, or tail excess. DP, EO, and calibration alter different parts of the confusion profile, and under heterogeneous costs those changes translate into distinct harm allocations. The resulting profiles separate rate alignment, predictive calibration, average harm, tail harm, and relational envy as different empirical objects.

### 8.5 Experiment 4: Fairness Interventions under Harm-Based Axioms

**Experimental setup.** Experiment 4 tests whether the surrogate–harm relationship changes when the intervention family is varied rather than only the statistical target. Using a shared linear logistic base learner, we compare **ERM**, **FairBalance**, **FairBatch**, **GroupDRO**, **FairMixup**, and **FairProjection**. These methods span unconstrained training, pre-processing, in-processing, distributionally robust training, data augmentation, and post-processing. All methods are evaluated on the same surrogate metrics and harm-based diagnostics under the fixed heterogeneous cost regime.

**Results and analysis.** Tables 6–7 show that varying the intervention family does not produce a monotone ordering between surrogate metrics and harm-based axioms. Methods with similar accuracy can occupy different positions in the surrogate–harm space, and improvements in DP gap, EO gap, or ECE do not determine the induced harm profile. Across tasks, the compared methods trade off mean harm, tail harm, maximum harm, and cost-stratum envy in different ways, indicating that algorithmic intervention choice changes the allocation of error-induced burdens rather than simply moving along a single fairness scale.

| Task | Model | Envy gap | | | |
|------|-------|------|-----|-----|-----|
| | | BASE | CAL | DP | EO |
| EMP | HGB | 5.3727 | 5.3754 | 5.8840 | 5.8060 |
| | LR | 6.3059 | 6.1411 | 6.0789 | 6.0727 |
| INC | HGB | 6.5272 | 6.0411 | 7.1476 | 6.3352 |
| | LR | 7.1331 | 7.2374 | 7.8536 | 5.9515 |
| COV | HGB | 13.0990 | 12.2058 | 10.6861 | 13.3671 |
| | LR | 10.7189 | 10.4490 | 12.9021 | 11.0700 |

(a)

| Task | Model | Max harm | | | |
|------|-------|------|-----|-----|-----|
| | | BASE | CAL | DP | EO |
| EMP | HGB | 212.8618 | 212.8618 | 212.8618 | 212.8618 |
| | LR | 212.8618 | 190.4720 | 212.8618 | 212.8618 |
| INC | HGB | 104.4634 | 104.4634 | 104.4634 | 104.4634 |
| | LR | 108.1216 | 108.1216 | 108.1216 | 108.1216 |
| COV | HGB | 168.8830 | 168.8830 | 150.4242 | 175.8929 |
| | LR | 168.8830 | 168.8830 | 168.8830 | 168.8830 |

(b)

| Task | Model | Mean harm | | | |
|------|-------|------|-----|-----|-----|
| | | BASE | CAL | DP | EO |
| EMP | HGB | 1.4630 | 1.4695 | 1.4760 | 1.4731 |
| | LR | 1.5837 | 1.5694 | 1.5591 | 1.5642 |
| INC | HGB | 1.0894 | 1.0714 | 1.0465 | 1.0205 |
| | LR | 1.3280 | 1.3592 | 1.3303 | 1.3254 |
| COV | HGB | 1.9602 | 1.9455 | 1.9584 | 2.0075 |
| | LR | 1.9811 | 1.9791 | 2.0001 | 1.9852 |

(c)

| Task | Model | Tail excess | | | |
|------|-------|------|-----|-----|-----|
| | | BASE | CAL | DP | EO |
| EMP | HGB | 1.2068 | 1.2128 | 1.2204 | 1.2149 |
| | LR | 1.2768 | 1.2623 | 1.2554 | 1.2594 |
| INC | HGB | 1.0257 | 1.0080 | 0.9827 | 0.9595 |
| | LR | 1.0875 | 1.1132 | 1.0908 | 1.0888 |
| COV | HGB | 1.2661 | 1.2516 | 1.2613 | 1.3028 |
| | LR | 1.2568 | 1.2567 | 1.2696 | 1.2605 |

(d)

Table 4: Harm-based metrics under fixed strong cost heterogeneity $(\gamma, \sigma) = (2, 1)$ for representative policies (BASE, CAL, DP, EO), averaged over three random seeds. Envy gap is computed as the maximum cost-stratum envy using a finer-grained partition into 80 cost strata.

| Task | Model | Accuracy | | | |
|------|-------|------|-----|-----|-----|
| | | BASE | CAL | DP | EO |
| EMP | HGB | 0.8146 | 0.8142 | 0.8137 | 0.8112 |
| | LR | 0.8023 | 0.8027 | 0.8027 | 0.8023 |
| INC | HGB | 0.8095 | 0.8084 | 0.8050 | 0.8121 |
| | LR | 0.7691 | 0.7691 | 0.7674 | 0.7744 |
| COV | HGB | 0.7090 | 0.7081 | 0.7069 | 0.7064 |
| | LR | 0.6949 | 0.6917 | 0.6905 | 0.6915 |

(a)

| Task | Model | DP gap | | | |
|------|-------|------|-----|-----|-----|
| | | BASE | CAL | DP | EO |
| EMP | HGB | 0.0171 | 0.0164 | 0.0103 | 0.0844 |
| | LR | 0.0140 | 0.0162 | 0.0138 | 0.0572 |
| INC | HGB | 0.0582 | 0.0587 | 0.0194 | 0.1376 |
| | LR | 0.0123 | 0.0106 | 0.0247 | 0.1678 |
| COV | HGB | 0.0145 | 0.0193 | 0.0171 | 0.1116 |
| | LR | 0.0319 | 0.0271 | 0.0249 | 0.1010 |

(b)

| Task | Model | ECE | | | |
|------|-------|------|-----|-----|-----|
| | | BASE | CAL | DP | EO |
| EMP | HGB | 0.0126 | 0.0099 | 0.0126 | 0.0126 |
| | LR | 0.0137 | 0.0150 | 0.0137 | 0.0137 |
| INC | HGB | 0.0158 | 0.0215 | 0.0158 | 0.0158 |
| | LR | 0.0206 | 0.0147 | 0.0206 | 0.0206 |
| COV | HGB | 0.0199 | 0.0237 | 0.0199 | 0.0199 |
| | LR | 0.0164 | 0.0242 | 0.0164 | 0.0164 |

(c)

| Task | Model | EO gap | | | |
|------|-------|------|-----|-----|-----|
| | | BASE | CAL | DP | EO |
| EMP | HGB | 0.0949 | 0.0937 | 0.0949 | 0.1384 |
| | LR | 0.0870 | 0.0863 | 0.0839 | 0.1071 |
| INC | HGB | 0.0333 | 0.0311 | 0.0998 | 0.0743 |
| | LR | 0.0806 | 0.0836 | 0.0778 | 0.1425 |
| COV | HGB | 0.0288 | 0.0354 | 0.0572 | 0.1267 |
| | LR | 0.0395 | 0.0348 | 0.0645 | 0.1188 |

(d)

Table 5: Accuracy and surrogate metrics for the same representative policies as Table 4, averaged over three random seeds.

## 9 Discussion

The preceding sections treat statistical fairness criteria as surrogate conditions for harm-allocation fairness. This discussion draws out the interpretation of that view: when surrogates align with harm, when they

| Task | Method | Accuracy | DP gap | EO gap | ECE |
|------|--------|----------|--------|--------|-----|
| EMP | ERM | 0.7991 | 0.0146 | 0.0783 | 0.0407 |
| | FairBalance | 0.7977 | 0.0177 | 0.0902 | 0.0429 |
| | FairBatch | 0.7929 | 0.0053 | 0.0802 | 0.0471 |
| | GroupDRO | 0.7955 | 0.0132 | 0.0791 | 0.0442 |
| | FairMixup | 0.8010 | 0.0151 | 0.0806 | 0.0269 |
| | FairProjection | 0.7992 | 0.0607 | 0.1055 | 0.0407 |
| INC | ERM | 0.7503 | 0.0041 | 0.0694 | 0.0807 |
| | FairBalance | 0.7656 | 0.0244 | 0.0930 | 0.0448 |
| | FairBatch | 0.7519 | 0.0386 | 0.0687 | 0.0801 |
| | GroupDRO | 0.7376 | 0.0246 | 0.0501 | 0.1012 |
| | FairMixup | 0.7602 | 0.0215 | 0.0631 | 0.0342 |
| | FairProjection | 0.7690 | 0.1505 | 0.1099 | 0.0807 |
| COV | ERM | 0.6719 | 0.0437 | 0.0532 | 0.0927 |
| | FairBalance | 0.6705 | 0.0373 | 0.0579 | 0.1176 |
| | FairBatch | 0.6836 | 0.0457 | 0.0786 | 0.0989 |
| | GroupDRO | 0.6448 | 0.1131 | 0.1307 | 0.1218 |
| | FairMixup | 0.6342 | 0.0225 | 0.0494 | 0.1110 |
| | FairProjection | 0.6856 | 0.1031 | 0.1200 | 0.0927 |

Table 6: Accuracy and surrogate metrics for six fairness methods, averaged over three random seeds.

| Task | Method | Mean harm | Max harm | Tail excess | Envy gap |
|------|--------|-----------|----------|-------------|----------|
| EMP | ERM | 1.5734 | 212.8618 | 1.2685 | 6.5496 |
| | FairBalance | 1.5806 | 190.4720 | 1.2702 | 5.9554 |
| | FairBatch | 1.5558 | 190.4720 | 1.2420 | 5.6830 |
| | GroupDRO | 1.5667 | 182.2880 | 1.2518 | 6.5817 |
| | FairMixup | 1.5717 | 212.8618 | 1.2675 | 5.9184 |
| | FairProjection | 1.5871 | 212.8618 | 1.2810 | 5.8274 |
| INC | ERM | 1.4685 | 108.1216 | 1.1559 | 6.0324 |
| | FairBalance | 1.3811 | 107.4177 | 1.0898 | 5.7819 |
| | FairBatch | 1.4879 | 148.2175 | 1.1738 | 7.0952 |
| | GroupDRO | 1.5973 | 149.0032 | 1.2650 | 6.0182 |
| | FairMixup | 1.4243 | 107.4177 | 1.1351 | 7.4050 |
| | FairProjection | 1.3353 | 108.1216 | 1.0623 | 6.7323 |
| COV | ERM | 2.0604 | 168.8830 | 1.2435 | 13.8251 |
| | FairBalance | 2.2162 | 163.3486 | 1.3606 | 12.1770 |
| | FairBatch | 2.0944 | 175.8929 | 1.2998 | 10.7925 |
| | GroupDRO | 2.2189 | 136.9072 | 1.3543 | 12.9062 |
| | FairMixup | 2.4023 | 131.5751 | 1.4496 | 11.0125 |
| | FairProjection | 2.0119 | 168.8830 | 1.2165 | 11.9692 |

Table 7: Harm-based metrics under fixed strong subjective-cost heterogeneity $(\gamma, \sigma) = (2, 1)$, averaged over three random seeds. Tail excess is measured relative to the ERM 90th-percentile harm threshold. Envy gap is computed as the maximum cost-stratum envy using a finer-grained partition into 80 cost strata.

diverge, how bounded rationality affects implementation, and what should be reported when fairness claims are made under heterogeneous error costs.

### 9.1 Surrogate alignment is conditional under subjective-cost heterogeneity

Statistical fairness metrics are meaningful constraints on observable decision behavior, but their connection to harm allocation depends on the structure of error costs. When harm weights are homogeneous or aligned

with the groups, labels, or score bins used by a metric, criteria such as DP, EO, and calibration can provide useful low-dimensional information about harm. The representation results identify this alignment condition: rate constraints become harm-informative when the coefficients translating FP/FN events into harm are comparable across the relevant partitions.

Cost heterogeneity weakens that surrogate relation. If individuals attach different burdens to the same FP/FN events, a policy can improve a statistical metric while redistributing errors toward individuals or cost strata for whom those errors are more severe. The experiments illustrate this conditional behavior rather than a universal ranking of metrics: DP often reduces cost-stratum envy in the tested setting, EO can increase it, and calibration tends to produce smaller mixed movements. The relevant lesson is that surrogate improvement should be interpreted through the induced harm allocation, especially when costs vary within groups.

### 9.2 Bounded rationality turns fairness into an approximation-floor problem

Harm-allocation fairness is implemented by intermediaries that observe costs imperfectly and choose from restricted policy classes. This makes exact axiom satisfaction less central than the residual violation attainable under institutional constraints. A nonzero envy or proportionality violation may arise even when the intermediary explicitly targets a harm-based objective, because the available policy class cannot induce every desired allocation and the observed cost representation may differ from the harm weights used for evaluation.

The lower-bound result isolates this mechanism in a minimal setting, while Experiment 2 shows its empirical form. Oracle cost information leaves a policy-class floor when simple thresholds cannot remove worst-stratum envy. Noisy or coarse cost information adds an information floor by changing which policy appears favorable on validation. These two components explain why stronger optimization of a fairness objective need not produce monotone improvement in realized harm fairness: the achievable region is determined jointly by cost information, policy expressiveness, and the validation procedure.

### 9.3 Statistical notions induce distinct, and sometimes conflicting, harm profiles

The harm diagnostics separate several distributive concerns. Mean harm describes aggregate burden; maximum harm and tail excess describe concentrated burden; envy describes relational grievance by asking whether one individual's assigned error burden looks worse than another's when evaluated under the same cost weights. These diagnostics connect the technical evaluation to familiar normative distinctions without requiring a single moral vocabulary to settle the problem. Aggregate welfare, protection of heavily burdened individuals, and comparative grievance can move in different directions under the same intervention.

Experiment 3 shows this separation in the profiles induced by DP, EO, and calibration. DP foregrounds selection-rate parity, EO foregrounds conditional error-rate parity, and calibration foregrounds score semantics. Each criterion regulates a different statistical object, so each can reshape the FP/FN distribution differently near the decision boundary. Under heterogeneous costs, those movements become different harm allocations: a policy can improve its targeted surrogate while leaving mean harm stable, changing tail exposure, or increasing relational envy. This explains why definition debates persist even when empirical facts are shared: different definitions prioritize different complaints.

### 9.4 Algorithmic diversity does not remove the structural trade-offs

The mismatch between surrogate metrics and harm allocation is not limited to a single post-processing template. Experiment 4 varies the intervention family by comparing unconstrained training, reweighting, batch-based training, distributionally robust optimization, data augmentation, and post-processing. These methods change the learned classifier or decision rule in different ways, but they still operate through cost-blind statistical objectives or constraints.

The resulting profiles show non-dominance rather than a stable method ranking. Methods with similar accuracy can differ sharply in mean harm, tail harm, maximum harm, and cost-stratum envy; a method that

improves one surrogate or one harm diagnostic need not improve the others. Algorithmic diversity therefore expands the attainable set of policies, but it does not collapse the surrogate–harm distinction. The choice among interventions remains a choice among different allocations of error-induced burdens.

### 9.5 Implications for evaluation and interpretation of "fair" ML systems

A fairness evaluation should report the statistical surrogate being targeted together with the harm model used to interpret its consequences. In the present framework, that means pairing DP gap, EO gap, calibration error, or accuracy with harm-facing diagnostics such as mean harm, tail excess, maximum harm, and envy. Reporting these quantities together prevents a rate-level improvement from being read as a complete fairness guarantee when the induced burden distribution moves differently.

The same evaluation should state the mediation setting: what cost information is available, how coarse or uncertain that information is, and which policy class is implementable. Under bounded rationality, fairness claims describe movements within an attainable region rather than convergence to an unconstrained ideal. A transparent deployment choice can then be expressed as a priority over that region, such as limiting relational grievance subject to a performance floor, controlling tail harm before optimizing mean harm, or using a statistical criterion as a constraint while auditing harm allocation directly.

### 9.6 Limitations and Future Work

The experiments use synthetic FP/FN cost weights to create controlled heterogeneity, so the results should be read as stress tests of surrogate–harm alignment rather than estimates of real-world damages or as a universal ranking of fairness methods. In deployment, cost representations may come from stakeholder elicitation, legal or policy principles, administrative categories, expert severity weights, or scenario-based sensitivity analysis. Each source raises measurement and legitimacy questions: elicited costs may be noisy or strategically reported, administrative categories may be too coarse, and legal or expert weights may encode institutional priorities rather than affected individuals' own valuations. The framework therefore treats costs as contestable harm representations rather than directly observed ground truth. The robustness analyses vary cost regimes and compare the cost-stratum proxy with individual-level envy to check whether the main patterns are artifacts of a single cost setting or diagnostic resolution.

The formal and empirical analysis focuses on realized-label FP/FN error burdens, which matches standard evaluation practice but does not exhaust the space of harm. Some realized FP/FN events may reflect avoidable model mistakes, while others arise from aleatoric or label uncertainty that even a Bayes-optimal policy cannot eliminate. Whether the latter should count as a fairness violation depends on the normative target: one may evaluate only avoidable error burdens, or evaluate the full allocation of realized burdens when affected individuals experience the same adverse outcome regardless of whether it was statistically avoidable. This distinction also separates model-mistake-induced harm from broader allocation-induced harm. The present experiments measure the former through FP/FN events; future work can extend the framework to expected-label accounting, uncertainty-aware harm decomposition, and multidimensional downstream burdens such as scrutiny, delay, stigma, or lost opportunity.

## 10 Conclusion

We studied ML fairness as a harm-allocation problem in which decision errors impose heterogeneous costs on affected individuals. This perspective treats Demographic Parity, Equalized Odds, and calibration as statistical surrogate criteria whose harm-based meaning depends on the alignment between rate-level constraints and the underlying distribution of error costs. The analysis showed that such alignment can hold under structured cost conditions, but can break under within- or across-group cost heterogeneity, producing violations of envy-freeness and proportionality-style protections even when statistical criteria are satisfied. Institutional bounded rationality—imperfect cost information together with restricted policy classes—can further create nonzero approximation floors for harm-based fairness. The experiments provide theorem-guided stress tests across ACS prediction tasks, cost regimes, post-processing rules, and fairness interventions, showing that surrogate improvements can coincide with, diverge from, or trade off against harm-allocation diagnostics.

These results support a more explicit evaluation practice: fairness claims should state the statistical criterion being targeted, the harm representation being used, and the information and policy constraints under which the resulting allocation is judged.

## Broader Impact Statement

This work provides a diagnostic framework for evaluating how ML decision systems allocate error-induced harms under heterogeneous cost representations. In high-stakes settings such as credit, employment, insurance, education, and public benefits, false positives and false negatives can impose uneven burdens across affected individuals. Making these burdens explicit can improve fairness audits by pairing standard statistical criteria, such as Demographic Parity, Equalized Odds, and calibration, with harm-allocation diagnostics that show who bears which errors under a given policy.

The framework can support more transparent governance by clarifying the object of evaluation. Rather than treating a rate-level improvement as a complete fairness guarantee, it asks whether the induced allocation of harms satisfies stated protections such as relational grievance or concentration of burden. This can help practitioners identify cases in which an intervention improves a statistical metric while shifting burdens toward individuals or groups for whom the relevant errors are more costly. It also encourages reporting results under multiple plausible cost representations, rather than presenting a single cost model as definitive.

The main risks concern institutional power, false precision, and value laundering. The actors who define, estimate, and operationalize harm costs may not be the same people who bear the resulting harms. Cost models can therefore shift normative authority from affected stakeholders to institutions, experts, or auditors. Numerical harm scores may also create false precision by making uncertain and contestable value judgments appear objective or settled. In addition, convenient cost assumptions could be selected to justify preferred policies while giving the appearance of principled fairness analysis.

Cost elicitation and measurement create further risks. Estimating preferences, vulnerabilities, legal interests, or downstream impacts may require sensitive information, introduce privacy burdens, or privilege the perspectives of stakeholders whose values are easier to measure. Coarse administrative categories may obscure within-group heterogeneity, while highly individualized estimates may increase surveillance or expose affected people to additional scrutiny. For these reasons, cost representations should be documented as contestable inputs, evaluated across plausible ranges or uncertainty sets, and developed with stakeholder participation where appropriate.

The framework should be used as an accountability tool rather than as a substitute for institutional judgment. Harm-allocation diagnostics can make trade-offs more visible, but they do not determine which harms matter most or who has authority to define them. Deployment decisions should therefore combine statistical metrics, harm-based diagnostics, sensitivity analyses, human oversight, and domain-specific legal or ethical constraints. The experiments in this paper are lightweight and do not materially increase computational or environmental burden.

## Reproducibility

All experiments in Section 8, as well as all tables reported in the paper, can be reproduced by running the single notebook `Price_of_Justice.ipynb` from start to finish. The notebook downloads and preprocesses the public Folktables ACS 2018 1-Year person survey, executes Experiments 1–4 as described in the paper—including model training, policy and algorithm construction, and evaluation—and reproduces the seed-averaged results in publication-style tabular form. It is designed to run both on Google Colab and in a local environment. Using the same random seeds and package versions yields identical tables, although minor numerical differences may arise across platforms or library versions because of nondeterminism in underlying numerical routines.

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

# A    Full Proofs for the Main Results

This appendix provides the proofs of the main theoretical results.

## A.1    Notation Used Throughout the Appendix

For an individual $i$, write

$$p_i^{FN} := \Pr_\pi(\text{FN} \mid i), \qquad p_i^{FP} := \Pr_\pi(\text{FP} \mid i).$$

The harm to individual $i$ from receiving $\ell$'s error lottery is

$$H_{i \leftarrow \ell}(\pi) := c_i^{FP} p_\ell^{FP} + c_i^{FN} p_\ell^{FN}.$$

The directed envy gap is

$$\text{EnvyGap}_{i,j}(\pi) := \left[ H_{i \leftarrow i}(\pi) - H_{i \leftarrow j}(\pi) \right]_+.$$

Aggregate envy statistics are obtained by aggregating these directed pairwise quantities, for example by taking maxima or averages over ordered pairs.

## A.2 Theorem 7.2: Representation Conditions

We use the following observation.

**Lemma A.1** (Cost-blind constraints and cost scaling)**.** *Fix $(\mathcal{D}, \pi)$, and let $M(\pi)$ be any statistical fairness metric defined as a functional of the induced law of $(A, Y, \hat{Y})$, and possibly $(A, Y, S)$. Then $M(\pi)$ is invariant to changes in the cost profile c.*

*If there exist $i \neq j$ such that*

$$(p_i^{FP}, p_i^{FN}) \neq (p_j^{FP}, p_j^{FN}),$$

*then, for any $B > 0$, there exists a cost profile c and an ordered pair $(a, b) \in \{(i, j), (j, i)\}$ such that*

$$\mathrm{EnvyGap}_{a,b}(\pi) \geq B,$$

*while $M(\pi)$ is unchanged.*

*Proof.* The value of $M(\pi)$ is determined by the distribution of predictions, labels, groups, and possibly scores. It does not depend on $c$.

Since the two error lotteries differ, at least one coordinate differs. Choose an ordered pair $(a, b)$ and an error type $r \in \{FP, FN\}$ such that

$$p_a^r - p_b^r = \Delta > 0.$$

Set $c_a^r = C$ and set the other coordinate of $c_a$ to zero. The remaining costs can be chosen arbitrarily without changing the following lower bound. Then

$$H_{a \leftarrow a}(\pi) - H_{a \leftarrow b}(\pi) = C(p_a^r - p_b^r) = C\Delta.$$

Taking $C := B/\Delta$ gives

$$\mathrm{EnvyGap}_{a,b}(\pi) \geq B.$$

The statistical metric remains unchanged because $(\mathcal{D}, \pi)$ is fixed. $\qquad\square$

*Proof.* Suppose a class of instances permits heterogeneous cost profiles and policies that induce distinct individual error lotteries. By Lemma A.1, any nonzero difference in individual error lotteries can be converted into an arbitrarily large envy gap by rescaling the cost of the affected error type. The statistical metric value is unchanged because the induced distribution of $(A, Y, \hat{Y})$, and possibly $(A, Y, S)$, is unchanged.

Thus, a statistical constraint can uniformly represent the harm-allocation axiom only if the instance class removes this scaling channel. This requires additional structure, such as equalized error lotteries at the relevant granularity or restrictions that collapse or align subjective costs. Without such restrictions, the statistical constraint and the harm-allocation axiom are not uniformly equivalent over the instance class. $\quad\square$

## A.3 Theorem 7.3: Counterexample Under Heterogeneous Costs

*Proof.* Fix $\eta \geq 0$ and let

$$M \in \{\mathrm{DP}, \mathrm{EO/EOp}, \mathrm{Calibration}\}.$$

Consider a single-group population, so $A$ is constant. Take two individuals $i \neq j$ with

$$Y_i = Y_j = 1.$$

Let the policy $\pi$ induce

$$p_i^{FN} = p, \qquad p_j^{FN} = q, \qquad p > q,$$

and set

$$p_i^{FP} = p_j^{FP} = 0.$$

Because there is only one group, the across-group gaps for DP and EO-type constraints are zero. For calibration, take the score to be $S \equiv 1$. Since all individuals in the construction have label $Y = 1$, groupwise calibration holds. Hence

$$M(\pi) = 0 \leq \eta.$$

Now choose costs

$$c_i^{FN} = C, \quad c_i^{FP} = 0, \qquad c_j^{FN} = 1, \quad c_j^{FP} = 0,$$

where $C > 0$. Then

$$\text{EnvyGap}_{i,j}(\pi) = \left[c_i^{FN}(p_i^{FN} - p_j^{FN})\right]_+ = C(p - q).$$

For any $B > 0$, choosing

$$C := \frac{B}{p - q}$$

gives

$$\text{EnvyGap}_{i,j}(\pi) \geq B, \qquad \text{Envy}^{\max}(\pi) \geq B,$$

while $M(\pi) \leq \eta$ still holds. $\qquad\square$

## A.4 Proposition 7.4: Statistical Impossibility as Axiom Conflict

*Proof.* Let

$$\mu_g := \Pr(Y = 1 \mid A = g)$$

denote the base rate in group $g$, and assume $\mu_a \neq \mu_b$.

First consider equalized odds. If EO holds with common true-positive and false-positive rates across groups, then for each group $g$,

$$\Pr(Y = 1 \mid \hat{Y} = 1, A = g) = \frac{\text{TPR} \cdot \mu_g}{\text{TPR} \cdot \mu_g + \text{FPR} \cdot (1 - \mu_g)}.$$

When $\text{TPR} \neq \text{FPR}$ and $\text{TPR}, \text{FPR} \in (0, 1)$, the right-hand side is strictly increasing in $\mu_g$. Hence unequal base rates imply unequal positive predictive values.

Now consider calibration at a common positive decision level. If $\hat{Y} = 1$ corresponds to a score bin $S = u$, and the score is calibrated within each group, then

$$\Pr(Y = 1 \mid \hat{Y} = 1, A = g) = \Pr(Y = 1 \mid S = u, A = g) = u$$

for each group $g$. Thus calibration equalizes the positive predictive value at that decision level.

Therefore, under unequal base rates, calibration and EO cannot both hold in the nondegenerate regime. Enforcing one of the two constraints leaves a nonzero gap in the other.

It remains to express this statistical gap as a harm-allocation gap. Suppose, for example, that the remaining EO violation is

$$\text{FNR}_a - \text{FNR}_b = \Delta > 0.$$

Choose individuals $i$ and $j$ with $A_i = a$, $A_j = b$, and $Y_i = Y_j = 1$, so that

$$p_i^{FN} - p_j^{FN} = \Delta.$$

Set

$$c_i^{FN} = C, \qquad c_i^{FP} = 0.$$

Then

$$\text{EnvyGap}_{i,j}(\pi) = C\Delta.$$

For any fixed tolerance, choosing $C$ sufficiently large makes the envy gap exceed that tolerance. Thus the statistical incompatibility can induce an arbitrarily large harm-axiom violation when subjective error costs are admissible. $\qquad\square$

### A.5 Theorem 7.5: A Lower Bound Under Bounded Rationality

*Proof.* Let $n = 2$ with labels

$$y_1 = 1, \qquad y_2 = 0.$$

Consider the restricted policy class $\Pi_{\text{simple}}$ in which both individuals receive the same positive prediction probability:

$$\Pr_\pi(\hat{Y} = 1 \mid 1) = \Pr_\pi(\hat{Y} = 1 \mid 2) =: \alpha, \qquad \alpha \in [0, 1].$$

Then

$$p_1^{FN} = 1 - \alpha, \qquad p_2^{FP} = \alpha.$$

The mediator observes proxy costs

$$\hat{c}_1 = \hat{c}_2 = (1, 1)$$

and faces the uncertainty set

$$\mathcal{C}(\hat{c}, \epsilon) = \left\{ c : c_i^{FP}, c_i^{FN} \in [1 - \epsilon, 1 + \epsilon] \text{ for } i = 1, 2 \right\}, \qquad \epsilon \in (0, 1).$$

For a fixed $\alpha$, the two directed envy gaps are

$$\text{EnvyGap}_{1,2}(\pi; c) = \left[ c_1^{FN}(1 - \alpha) - c_1^{FP}\alpha \right]_+,$$

and

$$\text{EnvyGap}_{2,1}(\pi; c) = \left[ c_2^{FP}\alpha - c_2^{FN}(1 - \alpha) \right]_+.$$

The first expression is maximized over $\mathcal{C}(\hat{c}, \epsilon)$ by setting

$$c_1^{FN} = 1 + \epsilon, \qquad c_1^{FP} = 1 - \epsilon,$$

so

$$\sup_{c \in \mathcal{C}(\hat{c}, \epsilon)} \text{EnvyGap}_{1,2}(\pi; c) = \left[ (1 + \epsilon)(1 - \alpha) - (1 - \epsilon)\alpha \right]_+ = \left[ 1 + \epsilon - 2\alpha \right]_+.$$

Similarly,

$$\sup_{c \in \mathcal{C}(\hat{c}, \epsilon)} \text{EnvyGap}_{2,1}(\pi; c) = \left[ (1 + \epsilon)\alpha - (1 - \epsilon)(1 - \alpha) \right]_+ = \left[ 2\alpha - (1 - \epsilon) \right]_+.$$

Therefore,

$$\sup_{c \in \mathcal{C}(\hat{c}, \epsilon)} \text{Envy}^{\max}(\pi; c) = \max\left\{ \left[ 1 + \epsilon - 2\alpha \right]_+, \left[ 2\alpha - (1 - \epsilon) \right]_+ \right\}.$$

The first term is decreasing in $\alpha$, and the second is increasing in $\alpha$. The minimum is obtained by balancing them:

$$1 + \epsilon - 2\alpha = 2\alpha - (1 - \epsilon),$$

which gives

$$\alpha = \frac{1}{2}.$$

At this value, both terms equal $\epsilon$. Hence

$$\inf_{\pi \in \Pi_{\text{simple}}} \sup_{c \in \mathcal{C}(\hat{c}, \epsilon)} \text{Envy}^{\max}(\pi; c) = \epsilon.$$

This establishes the lower bound. In this two-individual construction, the bound is tight. $\qquad \square$

## B Finite-Cost Lower Bounds

This section records finite-cost versions of the counterexample arguments. Throughout, assume

$$c_i^{FP}, c_i^{FN} \in [0, C_{\max}] \qquad \text{for all } i.$$

## B.1 Bounded-Cost Amplification

**Lemma B.1** (Bounded-cost amplification). *Fix $(\mathcal{D}, \pi)$ and two individuals $i \neq j$. Define*

$$\Delta_{i,j}^{FN} := p_i^{FN} - p_j^{FN}, \qquad \Delta_{i,j}^{FP} := p_i^{FP} - p_j^{FP}.$$

*Then*

$$\sup_{c_i^{FP}, c_i^{FN} \in [0, C_{\max}]} \mathrm{EnvyGap}_{i,j}(\pi) = C_{\max}\left((\Delta_{i,j}^{FP})_+ + (\Delta_{i,j}^{FN})_+\right).$$

*Consequently,*

$$\sup_{c \in [0, C_{\max}]^{2n}} \mathrm{Envy}^{\max}(\pi) = C_{\max} \max_{i,j}\left((\Delta_{i,j}^{FP})_+ + (\Delta_{i,j}^{FN})_+\right).$$

*Proof.* By definition,

$$\mathrm{EnvyGap}_{i,j}(\pi) = \left[c_i^{FP}(p_i^{FP} - p_j^{FP}) + c_i^{FN}(p_i^{FN} - p_j^{FN})\right]_+ = \left[c_i^{FP}\Delta_{i,j}^{FP} + c_i^{FN}\Delta_{i,j}^{FN}\right]_+.$$

The expression is linear in $c_i^{FP}$ and $c_i^{FN}$. A positive coefficient is maximized by setting the corresponding cost to $C_{\max}$, and a nonpositive coefficient is maximized by setting the corresponding cost to zero. Hence

$$\sup_{c_i^{FP}, c_i^{FN} \in [0, C_{\max}]} \mathrm{EnvyGap}_{i,j}(\pi) = C_{\max}\left((\Delta_{i,j}^{FP})_+ + (\Delta_{i,j}^{FN})_+\right).$$

Taking the maximum over ordered pairs gives the second claim. □

## B.2 A Finite-Cost Counterexample

**Proposition B.2** (Finite-cost surrogate failure). *Fix any $C_{\max} > 0$ and any tolerance $\eta \geq 0$. For each*

$$M \in \{\mathrm{DP}, \mathrm{EO/EOp}, \mathrm{Calibration}\},$$

*there exists an instance $(\mathcal{D}, c, \pi)$ such that*

$$M(\pi) \leq \eta, \qquad c_i^{FP}, c_i^{FN} \in [0, C_{\max}] \quad \text{for all } i,$$

*but*

$$\mathrm{Envy}^{\max}(\pi) \geq \frac{C_{\max}}{2}, \qquad \mathrm{Prop}^{\max}(\pi) \geq \frac{C_{\max}}{4}.$$

*Proof.* Consider a single-group population, so $A$ is constant. Take two individuals $i \neq j$ with

$$Y_i = Y_j = 1.$$

Let the policy $\pi$ induce

$$p_i^{FN} = \frac{3}{4}, \qquad p_j^{FN} = \frac{1}{4}, \qquad p_i^{FP} = p_j^{FP} = 0.$$

Then

$$p_i^{FN} - p_j^{FN} = \frac{1}{2}.$$

Since there is only one group, the across-group DP and EO-type gaps are zero. For calibration, take $S \equiv 1$. Because all individuals in the construction have $Y = 1$, calibration holds exactly. Therefore,

$$M(\pi) = 0 \leq \eta.$$

Choose bounded costs

$$c_i^{FN} = C_{\max}, \qquad c_i^{FP} = 0, \qquad c_j^{FN} = 0, \qquad c_j^{FP} = 0.$$

Then

$$\text{EnvyGap}_{i,j}(\pi) = \left[c_i^{FN}(p_i^{FN} - p_j^{FN})\right]_+ = \frac{C_{\max}}{2},$$

and hence

$$\text{Envy}^{\max}(\pi) \geq \frac{C_{\max}}{2}.$$

For proportionality, the two-individual proportional-share benchmark for individual $i$ is

$$\frac{1}{2}\Big(H_{i \leftarrow i}(\pi) + H_{i \leftarrow j}(\pi)\Big).$$

Thus

$$\text{PropExcess}_i(\pi) = \left[H_{i \leftarrow i}(\pi) - \frac{1}{2}\Big(H_{i \leftarrow i}(\pi) + H_{i \leftarrow j}(\pi)\Big)\right]_+ = \frac{1}{2}\Big[H_{i \leftarrow i}(\pi) - H_{i \leftarrow j}(\pi)\Big]_+.$$

Since the final term is $\text{EnvyGap}_{i,j}(\pi)$,

$$\text{PropExcess}_i(\pi) = \frac{1}{2}\,\text{EnvyGap}_{i,j}(\pi) = \frac{C_{\max}}{4}.$$

Therefore,

$$\text{Prop}^{\max}(\pi) \geq \frac{C_{\max}}{4}.$$

$\square$

## C  Generalized Harm Models

### C.1  Event Bundles and Harm Functionals

Let $\mathcal{K}$ be a finite index set of harmful events. The FP/FN model is the special case

$$\mathcal{K} = \{\text{FP}, \text{FN}\}.$$

More generally, $\mathcal{K}$ may include any policy-dependent harmful event, such as audits, delays, additional scrutiny, or other adverse interventions.

For each individual $\ell$, a policy $\pi$ induces an event-probability bundle

$$p_\ell \in [0,1]^{|\mathcal{K}|}, \qquad p_{\ell,k} := \Pr_\pi(E_k \mid \ell) \quad (k \in \mathcal{K}).$$

Individual $i$ evaluates $\ell$'s bundle through a monotone harm functional

$$u_i : [0,1]^{|\mathcal{K}|} \to \mathbb{R}_+, \qquad H_{i \leftarrow \ell}(\pi) := u_i(p_\ell).$$

The directed envy gap becomes

$$\text{EnvyGap}_{i,j}(\pi) := \big[u_i(p_i) - u_i(p_j)\big]_+.$$

### C.2  Single-Coordinate Sensitivity

The main constructions require only that a policy can induce a gap in some harmful-event coordinate and that the admissible harm model can assign value to that coordinate. This is captured by the following condition.

**Coordinate-richness condition.** For every $k \in \mathcal{K}$ and every $\theta \geq 0$, the admissible family of harm functionals contains

$$u_{i,k,\theta}(p) := \theta p_k.$$

The additive FP/FN model satisfies this condition. So does the additive multi-event model

$$u_i(p) = \sum_{k \in \mathcal{K}} c_{i,k} p_k, \qquad c_{i,k} \geq 0.$$

**Proposition C.1** (Generalized surrogate failure)**.** *Assume the coordinate-richness condition. Fix $(\mathcal{D}, \pi)$ and suppose there exist individuals $i \neq j$ and an event coordinate $k \in \mathcal{K}$ such that*

$$p_{i,k} \neq p_{j,k}.$$

*Then, for any $B > 0$, there exists an admissible harm functional $u_i$ such that*

$$\mathrm{EnvyGap}_{i,j}(\pi) \geq B$$

*after possibly swapping the order of $(i, j)$. Any statistical metric depending only on the induced law of $(A, Y, \hat{Y})$, and possibly scores $S$, is unchanged.*

*Proof.* Choose an ordering of $(i, j)$ such that

$$\Delta_k := p_{i,k} - p_{j,k} > 0.$$

By coordinate richness, take

$$u_i(p) = u_{i,k,\theta}(p) = \theta p_k$$

with

$$\theta := \frac{B}{\Delta_k}.$$

Then

$$u_i(p_i) - u_i(p_j) = \theta(p_{i,k} - p_{j,k}) = B,$$

and hence

$$\mathrm{EnvyGap}_{i,j}(\pi) = B.$$

The statistical metric is unchanged because $(\mathcal{D}, \pi)$, and therefore the induced distribution of predictions, labels, groups, and scores, has not been modified. $\square$

## C.3 Bounded-Rationality Floors

The bounded-rationality lower bound also extends to event-bundle harms. Let $\Pi_{\mathrm{simple}}$ be a restricted policy class, and suppose there exist $\delta > 0$ and an event coordinate $k \in \mathcal{K}$ such that every $\pi \in \Pi_{\mathrm{simple}}$ leaves some ordered pair $(i, j)$ with

$$p_{i,k} - p_{j,k} \geq \delta.$$

Assume the mediator only knows a proxy sensitivity $\hat{\theta}$, while the true sensitivity can lie in

$$[\hat{\theta}, \hat{\theta} + \epsilon], \qquad \epsilon > 0,$$

and that $u_{i,k,\theta}(p) = \theta p_k$ is admissible for all $\theta$ in this interval.

**Proposition C.2** (Generalized minimax floor)**.** *Under the conditions above,*

$$\inf_{\pi \in \Pi_{\mathrm{simple}}} \sup_{\theta \in [\hat{\theta}, \hat{\theta} + \epsilon]} \mathrm{EnvyGap}_{i,j}(\pi) \geq (\hat{\theta} + \epsilon)\delta \geq \epsilon\delta.$$

*Proof.* For any $\pi \in \Pi_{\mathrm{simple}}$, choose an ordered pair $(i, j)$ satisfying

$$p_{i,k} - p_{j,k} \geq \delta.$$

For $u_{i,k,\theta}(p) = \theta p_k$,

$$\mathrm{EnvyGap}_{i,j}(\pi) = \left[\theta(p_{i,k} - p_{j,k})\right]_+.$$

The supremum over $\theta \in [\hat{\theta}, \hat{\theta} + \epsilon]$ is attained at $\theta = \hat{\theta} + \epsilon$, so

$$\sup_{\theta \in [\hat{\theta}, \hat{\theta} + \epsilon]} \mathrm{EnvyGap}_{i,j}(\pi) \geq (\hat{\theta} + \epsilon)\delta.$$

Taking the infimum over $\pi \in \Pi_{\mathrm{simple}}$ gives the result. $\square$

# D   Auxiliary Lemmas and Tightness Notes

This section records two auxiliary facts used in the appendix: a two-agent identity connecting proportionality excess to directed envy, and the tightness of the bounded-rationality floor constructed in Theorem 7.5.

## D.1   Two-Agent Proportionality Identity

**Lemma D.1** (Two-agent proportionality and directed envy). *Let $n = 2$. For individual $i$, let $j \neq i$ and define the proportional-share benchmark as*

$$\mathrm{PS}_i(\pi) := \frac{1}{2}\Big(H_{i \leftarrow i}(\pi) + H_{i \leftarrow j}(\pi)\Big).$$

*Then*

$$\mathrm{PropExcess}_i(\pi) := \big[H_{i \leftarrow i}(\pi) - \mathrm{PS}_i(\pi)\big]_+ = \frac{1}{2}\,\mathrm{EnvyGap}_{i,j}(\pi).$$

*Consequently,*

$$\mathrm{Prop}^{\max}(\pi) = \frac{1}{2}\,\mathrm{Envy}^{\max}(\pi)$$

*on any two-individual instance.*

*Proof.* Expanding the definition gives

$$\mathrm{PropExcess}_i(\pi) = \left[H_{i \leftarrow i}(\pi) - \frac{1}{2}\Big(H_{i \leftarrow i}(\pi) + H_{i \leftarrow j}(\pi)\Big)\right]_+.$$

Hence

$$\mathrm{PropExcess}_i(\pi) = \frac{1}{2}\big[H_{i \leftarrow i}(\pi) - H_{i \leftarrow j}(\pi)\big]_+ = \frac{1}{2}\,\mathrm{EnvyGap}_{i,j}(\pi).$$

Taking maxima over the two individuals gives

$$\mathrm{Prop}^{\max}(\pi) = \frac{1}{2}\,\mathrm{Envy}^{\max}(\pi).$$

$\square$

## D.2   Tightness of the Bounded-Rationality Floor

**Proposition D.2** (Tightness of the bounded-rationality construction). *For the two-individual instance used in Theorem 7.5,*

$$\inf_{\pi \in \Pi_{\mathrm{simple}}} \sup_{c \in \mathcal{C}(\hat{c}, \epsilon)} \mathrm{Envy}^{\max}(\pi; c) = \epsilon.$$

*Proof.* In the construction of Theorem 7.5, policies in $\Pi_{\mathrm{simple}}$ are indexed by

$$\alpha \in [0, 1],$$

where both individuals receive positive prediction probability $\alpha$. The robust directed envy terms are

$$\big[1 + \epsilon - 2\alpha\big]_+, \qquad \big[2\alpha - (1 - \epsilon)\big]_+.$$

Therefore,

$$\sup_{c \in \mathcal{C}(\hat{c}, \epsilon)} \mathrm{Envy}^{\max}(\pi; c) = \max\Big\{\big[1 + \epsilon - 2\alpha\big]_+, \big[2\alpha - (1 - \epsilon)\big]_+\Big\}.$$

The first term is decreasing in $\alpha$, while the second is increasing in $\alpha$. They are equal at

$$\alpha = \frac{1}{2},$$

and both equal $\epsilon$ there. Hence the minimax value is at least $\epsilon$ and is attained by the policy with $\alpha = 1/2$. For this policy,

$$\text{EnvyGap}_{1,2}(\pi; c) = \left[ \frac{1}{2}(c_1^{FN} - c_1^{FP}) \right]_+, \qquad \text{EnvyGap}_{2,1}(\pi; c) = \left[ \frac{1}{2}(c_2^{FP} - c_2^{FN}) \right]_+.$$

Since

$$c_i^{FP}, c_i^{FN} \in [1 - \epsilon, 1 + \epsilon],$$

each directed gap is at most

$$\frac{1}{2}\big((1 + \epsilon) - (1 - \epsilon)\big) = \epsilon.$$

Thus

$$\sup_{c \in \mathcal{C}(\hat{c}, \epsilon)} \text{Envy}^{\max}(\pi; c) \le \epsilon.$$

Combining the lower and upper bounds gives the claimed equality. $\square$

## E  Robustness Analyses for Proxy Envy and Cost Heterogeneity

This appendix reports two robustness checks for the empirical harm diagnostics used in Section 8. We first examine whether the cost-stratum envy proxy is sensitive to the number of strata $K$. We then vary the heterogeneity-strength parameter $\gamma$ to test whether surrogate–axiom gaps appear only under extreme cost heterogeneity.

### E.1  $K$-sweep Stability of the Cost-Stratum Envy Proxy

**Experimental setup.**  We evaluate the sensitivity of the cost-stratum envy proxy to the discretization level $K$. Using the same ACS tasks, model classes, random seeds, group attribute, and surrogate policy families as in the main experiments, we recompute the proxy under $K \in \{20, 40, 80, 160, 320\}$, with $K = 80$ as the reference value. For each setting, individuals are partitioned into cost strata according to their relative FP/FN costs, and maximum type-level envy is computed for each candidate policy. Stability is measured by comparing the policy-level envy vector at each $K$ against the vector at $K = 80$, using median Spearman rank correlation and Top-1 agreement of the envy-minimizing policy. We report results under both the main heterogeneity setting (HETERO) and the correlation-sensitivity setting (HETERO-$\rho$), excluding degenerate cases in which the reference envy vector is constant.

**Results and analysis.**  Table 8 shows that the proxy is reasonably stable around the reference choice $K = 80$, though stability varies by surrogate family and scenario. EO is the most stable overall, especially under the $\rho$-sweep, where rank correlations and Top-1 agreement remain high for nearby discretizations. DP shows moderate rank stability but weaker Top-1 agreement, indicating that the identity of the envy-minimizing policy can be sensitive even when the broader ranking is partially preserved. CAL falls between these cases, with better stability at intermediate values of $K$ and weaker stability under very coarse or very fine discretization. These results support using the cost-stratum proxy as a comparative diagnostic rather than as a point estimate tied to a single binning choice.

### E.2  $\gamma$-sweep Sensitivity to the Strength of Cost Heterogeneity

**Experimental setup.**  We next vary the heterogeneity-strength parameter $\gamma$, which controls group-dependent cost scaling, over

$$\gamma \in \{1.0, 1.25, 1.5, 2.0, 3.0, 5.0\},$$

while holding the remaining cost parameters fixed. For each ACS task, model class, seed, and surrogate family, we use the same candidate-policy generation pipeline as in the main experiments. At each $\gamma$, we compute an envy-based axiom gap: within a surrogate family, we select the surrogate-optimal policy as

Table 8: Stability of the cost-stratum envy proxy with respect to the number of strata $K$, measured relative to the reference choice $K = 80$. We report (i) the median Spearman rank correlation between the policy-level envy vector at $K$ and that at $K = 80$, and (ii) the Top-1 agreement rate (i.e., whether the envy-minimizing policy matches that selected at $K = 80$). $N$ denotes the number of non-degenerate setting instances included in the aggregation.

| Scenario | Policy | $N$ | Spearman rank correlation: $K{=}80$ | | | | Top-1 agreement: $K{=}80$ | | | |
|---|---|---|---|---|---|---|---|---|---|---|
| | | | $K{=}20$ | $K{=}40$ | $K{=}160$ | $K{=}320$ | $K{=}20$ | $K{=}40$ | $K{=}160$ | $K{=}320$ |
| | DP | 18 | 0.6429 | 0.7679 | 0.7679 | 0.4821 | 0.3333 | 0.6111 | 0.3333 | 0.1667 |
| HETERO | EO | 18 | 0.5000 | 0.5000 | 1.0000 | 0.9330 | 0.5556 | 0.4444 | 0.6111 | 0.6111 |
| | CAL | 15 | 0.8030 | 0.7456 | 0.3627 | 0.2962 | 0.5333 | 0.4000 | 0.4667 | 0.4667 |
| | DP | 90 | 0.5357 | 0.6099 | 0.6964 | 0.6726 | 0.3556 | 0.4444 | 0.3778 | 0.4000 |
| HETERO-$\rho$ | EO | 90 | 0.5000 | 1.0000 | 1.0000 | 0.8660 | 0.5556 | 0.6222 | 0.7333 | 0.6889 |
| | CAL | 68 | 0.3953 | 0.7906 | 0.7906 | 0.4326 | 0.4265 | 0.5735 | 0.4853 | 0.4118 |

the lowest-error policy in that family, and compare its envy to the minimum envy attainable over the full candidate set. Formally,

$$\mathrm{Gap}(\gamma) = E(\pi_{\mathrm{sur}}(\gamma)) - \min_{\pi \in \Pi} E(\pi; \gamma),$$

where $E(\cdot)$ is the cost-stratum envy proxy computed with $K = 80$, and $\Pi$ denotes the full set of candidate policies. We report the median and interquartile range across all task–model–seed settings.

**Results and analysis.** Table 9 and Figure 1 show that the axiom gap is already nonzero under mild heterogeneity and generally increases as $\gamma$ grows. This pattern appears across DP, EO, and CAL, indicating that surrogate-optimal policy selection can be misaligned with envy minimization even when cost disparities are not extreme. Larger values of $\gamma$ also produce wider interquartile ranges, suggesting that the strength of misalignment becomes more setting-dependent as cost heterogeneity increases. Thus, the main empirical pattern is not driven solely by an extreme cost scale; stronger heterogeneity amplifies the gap, but the gap is already visible in milder regimes.

Table 9: Robustness to the strength of cost heterogeneity $\gamma$. Top: median axiom gap. Bottom: interquartile range (25th–75th percentiles).

(a) Median axiom gap $E(\pi_{\mathrm{sur}}) - \min_{\pi} E(\pi)$.

| | $\gamma{=}1.0$ | $\gamma{=}1.25$ | $\gamma{=}1.5$ | $\gamma{=}2.0$ | $\gamma{=}3.0$ | $\gamma{=}5.0$ |
|---|---|---|---|---|---|---|
| DP | 0.3026 | 0.5478 | 0.8002 | 3.2951 | 3.1256 | 27.8322 |
| EO | 0.4087 | 0.5630 | 0.7149 | 3.1453 | 3.5498 | 17.0974 |
| CAL | 0.2758 | 0.5383 | 0.5223 | 3.5701 | 2.7258 | 16.1681 |

(b) IQR (25th–75th pct.) of the axiom gap.

| | $\gamma{=}1.0$ | $\gamma{=}1.25$ | $\gamma{=}1.5$ | $\gamma{=}2.0$ | $\gamma{=}3.0$ | $\gamma{=}5.0$ |
|---|---|---|---|---|---|---|
| DP | 0.1177–0.8750 | 0.2388–1.2873 | 0.3675–1.6872 | 2.8084–4.2489 | 1.8608–5.7724 | 15.9506–46.6524 |
| EO | 0.1689–0.7194 | 0.3591–1.1283 | 0.3649–1.2010 | 1.6267–5.0086 | 1.3076–5.7085 | 5.7565–26.8462 |
| CAL | 0.1061–0.4944 | 0.0941–0.8058 | 0.2607–1.5470 | 2.8042–4.2714 | 0.4738–5.7549 | 5.0628–38.3248 |

### E.3 Exact Individual Envy and Cost-Stratum Proxy Validation

**Experimental setup.** We validate the cost-stratum envy proxy against exact individual-level envy on smaller evaluation subsets where exact comparisons are tractable. For each ACS task, model class, random seed, and candidate-policy family used in the main experiments, we sample evaluation subsets of size 250, 500, and 1000, with five repetitions per setting. On each subset, we compute exact individual envy from

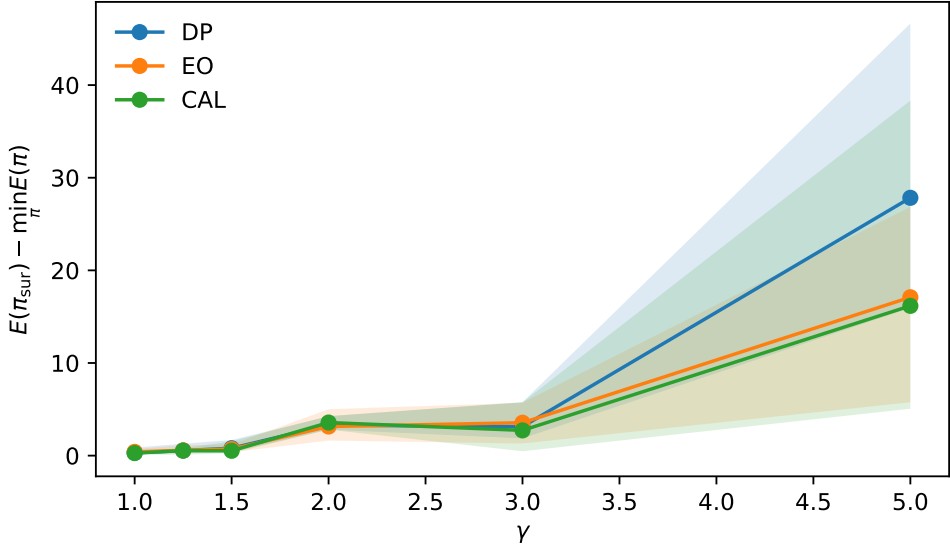

Figure 1: Trend of the axiom gap as $\gamma$ increases (median with IQR band).

realized FP/FN bundles and compare it with the cost-stratum proxy computed using the default $K = 80$ strata. Because the proxy is used as a comparative diagnostic rather than as a pointwise estimator of every pairwise envy relation, we evaluate whether it preserves policy-level ordering. Specifically, Table 10 reports the median Spearman rank correlation between the proxy and two exact-envy summaries: average individual envy and the fraction of envious individuals. We also report Top-1 agreement, defined as whether the proxy and exact average envy select the same minimum-envy policy within the comparison set.

**Results and analysis.**  Table 10 shows that the cost-stratum proxy preserves a meaningful policy-level signal, especially for average envy and the fraction of envious individuals. Across the full candidate set, the median rank correlation with exact average envy increases from 0.6298 at subset size 250 to 0.7154 at subset size 1000, and the corresponding correlation with the envious-individual fraction remains consistently positive. Within surrogate families, EO exhibits the strongest agreement, with stable rank correlations of 0.8660 across all subset sizes and high Top-1 agreement. DP shows moderate alignment, particularly for exact average envy and envy fraction. CAL yields fewer non-degenerate rank-correlation cases because several calibration-mixing policies induce nearly identical classifications under the fixed threshold, but its non-degenerate comparisons still show positive average-envy alignment at larger subset sizes. Overall, these results support the use of the cost-stratum proxy as a scalable comparative diagnostic for policy-level harm allocation, while avoiding the stronger claim that it exactly reproduces individual pairwise envy.

### E.4   Sensitivity to Cost Heterogeneity Regimes

**Experimental setup.**  We evaluate whether the surrogate–axiom gaps observed in the main experiments depend on a single cost specification. Using the same ACS tasks, model classes, random seeds, group attribute, and candidate-policy pipeline as in the main experiments, we vary the cost-heterogeneity parameters over a grid

$$\gamma \in \{1.0, 2.0, 3.0\}, \qquad \sigma \in \{0.5, 1.0, 1.5\},$$

with $\rho = -1$ fixed. The parameter $\gamma$ controls between-group cost scaling, while $\sigma$ controls within-group FP/FN cost heterogeneity. For each task–model–seed setting, candidate policies are generated once and then re-evaluated under each cost regime using the cost-stratum envy proxy with $K = 80$. For each surrogate family $F \in \{DP, EO, CAL\}$, we select the lowest-error policy within that family and compare its envy to the minimum envy attainable over the full candidate set $\Pi$, including the baseline and all surrogate-family

Table 10: Validation of the cost-stratum envy proxy against exact individual-level envy. $N$ denotes the number of evaluated setting instances, and $N_{\text{valid}}$ denotes the number of non-degenerate instances for which the Spearman correlation with exact average envy is defined.

| Comparison | Subset size | $N$ | $N_{\text{valid}}$ | Spearman Avg. | Spearman Frac. | Top-1 Avg. |
|---|---|---|---|---|---|---|
| ALL | 250 | 90 | 90 | 0.6298 | 0.6205 | 0.5778 |
| | 500 | 90 | 90 | 0.6893 | 0.6721 | 0.4111 |
| | 1000 | 90 | 90 | 0.7154 | 0.6868 | 0.4000 |
| DP | 250 | 90 | 89 | 0.6910 | 0.5853 | 0.5667 |
| | 500 | 90 | 90 | 0.7759 | 0.7042 | 0.4444 |
| | 1000 | 90 | 90 | 0.7085 | 0.7175 | 0.4000 |
| EO | 250 | 90 | 85 | 0.8660 | 0.8660 | 0.8778 |
| | 500 | 90 | 89 | 0.8660 | 0.8660 | 0.7889 |
| | 1000 | 90 | 90 | 0.8660 | 0.8660 | 0.7444 |
| CAL | 250 | 90 | 10 | -0.3064 | -0.2182 | 0.9444 |
| | 500 | 90 | 13 | 0.7255 | 0.3227 | 0.9778 |
| | 1000 | 90 | 20 | 0.6698 | 0.3847 | 0.9111 |

candidates:

$$\text{Gap}_F(\gamma, \sigma) = E(\pi_F; \gamma, \sigma) - \min_{\pi \in \Pi} E(\pi; \gamma, \sigma),$$

where $E(\cdot)$ denotes the cost-stratum envy proxy. Table 11 reports the median gap across task–model–seed settings.

**Results and analysis.** Table 11 shows that the surrogate–axiom gap is not an artifact of the default cost setting $(\gamma, \sigma) = (2, 1)$. The median gap is positive for every surrogate family in every regime, including the mildest heterogeneity setting. The magnitude of the gap also increases as either between-group scaling or within-group FP/FN heterogeneity becomes stronger, which is consistent with the theoretical claim that statistical surrogate constraints do not directly control harm allocations under heterogeneous costs. The numerical magnitude varies across DP, EO, and CAL, but the table should not be read as a universal ranking of fairness criteria. Its purpose is to show that the main empirical pattern persists across multiple cost regimes: accuracy-selected policies within statistical surrogate families can remain separated from the envy-minimizing policy even when the assumed cost model is varied.

Table 11: Sensitivity of the surrogate–axiom gap to cost heterogeneity regimes. Each entry reports the median envy-based axiom gap across task–model–seed settings.

| | $\gamma = 1.0$ | | | $\gamma = 2.0$ | | | $\gamma = 3.0$ | | |
|---|---|---|---|---|---|---|---|---|---|
| Family | $\sigma = 0.5$ | $\sigma = 1.0$ | $\sigma = 1.5$ | $\sigma = 0.5$ | $\sigma = 1.0$ | $\sigma = 1.5$ | $\sigma = 0.5$ | $\sigma = 1.0$ | $\sigma = 1.5$ |
| DP | 0.1232 | 0.5893 | 2.4849 | 0.3267 | 1.4222 | 6.2132 | 0.8909 | 3.7552 | 16.6901 |
| EO | 0.1643 | 0.7182 | 3.5141 | 0.4129 | 1.7380 | 8.3099 | 1.1100 | 4.6455 | 22.1316 |
| CAL | 0.1632 | 0.7987 | 3.7641 | 0.3953 | 2.0048 | 8.9632 | 1.0567 | 5.3885 | 23.8979 |

## F    Discussion of Modeling Scope

This section clarifies the scope of the theoretical results. The separation results rely on two ingredients: a policy induces heterogeneous harm-relevant bundles, and the admissible value class allows individuals to weight at least one bundle coordinate differently. In the main text, the bundle is

$$(p_i^{FP}, p_i^{FN}),$$

but the same logic applies to richer policy-dependent harm events, as discussed in Appendix C. The results are therefore policy-level statements about the relationship between statistical constraints and harm-allocation axioms, not claims about a particular learning algorithm, optimization procedure, or application domain.

The results do not imply that statistical fairness metrics can never align with harm-based axioms. Such alignment can occur under additional structure. For example, if individuals share a common value scale, then controlling error rates can also control harm. Likewise, if a policy equalizes the relevant harm bundle itself, then envy violations vanish at the relevant comparison level. The negative results identify the assumptions needed for statistical definitions to serve as faithful surrogates, rather than denying the usefulness of such surrogates in restricted regimes.

Several modeling choices are used for clarity rather than necessity. Linear FP/FN costs provide the simplest measurable harm model, but the arguments only require directional sensitivity to some policy-dependent harmful event. Two-agent constructions are proof devices; the same mechanisms can be embedded in larger populations by holding other individuals' bundles fixed or assigning them neutral costs. Single-group counterexamples show that harm-axiom violations can arise even without group structure; with group structure, the same reasoning can be applied within or across groups depending on the statistical notion under study.

The bounded-rationality lower bounds capture residual axiom violation under limited information and restricted policy expressiveness. Formally, they study quantities of the form

$$\inf_{\pi \in \Pi_{\text{simple}}} \sup_{c \in \mathcal{C}(\hat{c}, \epsilon)} V(\pi; c),$$

where $V(\pi; c)$ is a harm-axiom violation score, $\Pi_{\text{simple}}$ is a restricted policy class, and $\mathcal{C}(\hat{c}, \epsilon)$ represents uncertainty around proxy costs. The residual arises when the mediator cannot both observe the true subjective costs and implement the allocation needed for all feasible realizations of those costs. Appendix D records tightness for the canonical two-agent construction.

