# OpenReview forum: "The Price of Justice in Machine Learning: Fair Division with Subjective Value under Bounded Rationality"
_TMLR — Rejected by TMLR_

### Review · Reviewer_8E3C · 2026-03-25

**Summary Of Contributions:**

The paper argues that common statistical fairness notions such as Demographic Parity, Equalized Odds, and Calibration should be treated as surrogates rather than fairness itself, because they constrain rates and may fail to capture how burdens are distributed across individuals. The core proposal is to model fairness through individual error costs, define a harm-allocation vector induced by a decision policy, and evaluate that allocation using fair-division axioms such as envy-freeness and proportionality. It further studies bounded rationality, where the mediator has only noisy or coarse access to individuals’ costs and is restricted to simple policy classes, and argues that this can create an irreducible approximation floor for fairness. The paper’s main theoretical contribution is to characterize when statistical metrics can validly stand in for harm-based fairness, construct counterexamples showing structural surrogate failure under cost heterogeneity, and reinterpret classic fairness incompatibility results as conflicts among underlying harm-allocation principles.

**Audience:**

Yes

**Audience Explanation:**

This paper argues that common statistical fairness criteria may fail to capture how burdens are distributed across individuals. The topic is timely and relevant to researchers working on fairness.

**Claims And Evidence:**

Yes

**Claims Explanation:**

1. While the paper discusses bounded rationality through noisy or coarse proxies of subjective costs, the more fundamental practical difficulty is that such individual-specific FP/FN costs may be unavailable. There is no clear elicitation or measurement protocol for defining these individual costs $c_i$.

2. The paper relies heavily on the cost-stratum envy proxy in its empirical evaluation, yet its reliability is not fully validated. While the proxy is motivated as a scalable approximation, the paper does not provide direct evidence showing how well it matches exact individual-level envy. It would strengthen the empirical support substantially if the authors included an additional experiment on a small toy dataset, where exact envy can be computed, to compare the proxy against the true envy metric.

3. Theorem 7.1 establishes that surrogate validity requires additional restrictions on subjective costs, but the paper mainly focuses on the negative result that DP/EO can fail as surrogates for harm-based fairness. However, Experiment 1 seems to show that DP can still consistently improve envy, suggesting DP can be the surrogates. Could the authors clarify under what assumptions a statistical fairness metric can reliably serve as a surrogate for harm-based fairness?

4. The experiments appear to rely largely on the same underlying simulated cost setting (2,1). Could the author provide more experiments in different settings?

**Requested Changes:**

I believe the paper would be significantly strengthened by addressing the concerns above. In particular, the authors should focus on providing stronger experimental evidence for the accuracy of the proxy envy and for the generality of the findings across different cost settings.

---

> ### Author Response · Authors · 2026-05-01
> **Response to Reviewer 8E3C**
>
> Dear Reviewer 8E3C,
>
> Thank you very much for your thoughtful and constructive review. We appreciate your recognition of the paper’s central aim: to examine when standard statistical fairness criteria can serve as surrogate constraints for the allocation of individualized error-induced harms. Your comments were especially helpful in strengthening the cost model, the empirical diagnostics, and the interpretation of the DP results.
>
> You raised an important concern about the practical availability of individual false-positive and false-negative costs. We revised Sections 4 and 6 to clarify how these quantities should be understood. Section 4 now defines them as harm weights that may come from stakeholder elicitation, welfare-loss estimates, legal or policy principles, expert assessment, or stakeholder-informed severity judgments. Section 6 further clarifies that institutions usually operate with noisy, coarse, or proxy-based cost representations rather than perfectly observed individual costs. The bounded-rationality framework is therefore presented as a model of this limitation, not as a claim that cost elicitation is straightforward.
>
> Your concern about the cost-stratum envy proxy also led to a substantive revision. Section 5 now makes the proxy status explicit, and Appendix E.3 adds a new validation analysis comparing the proxy with exact individual-level envy on smaller evaluation subsets. We compare them at the policy-ranking level using rank correlations, agreement with the fraction of envious individuals, and Top-1 agreement. The results support using cost-stratum envy as a scalable comparative diagnostic, while we avoid claiming that it exactly reproduces all pairwise envy relations.
>
> We also revised the interpretation of the empirical DP results. As you noted, DP-selected policies sometimes reduce envy in our experiments. The revised manuscript now explains this in Sections 7.1, 8.2, and 9.1. We do not claim that DP, EO, or Calibration are useless or always harmful. Rather, their validity as harm-based surrogates depends on structural alignment between statistical partitions and harm weights. Section 7.1 now includes a positive surrogate-alignment result, and the empirical discussion treats the DP improvement as a setting-specific case of partial alignment rather than as a contradiction of the theoretical results.
>
> In response to your request for broader cost-regime sensitivity analysis, we added Appendix E.4. This new analysis varies both the between-group cost-scaling parameter $\gamma$ and the within-group FP/FN heterogeneity parameter $\sigma$ over a grid of regimes, and recomputes the surrogate–axiom gap for DP, EO, and CAL policy families. The results support the conclusion that the observed surrogate–harm misalignment is not an artifact of the default cost specification.
>
> Thank you again for these helpful suggestions. They led us to clarify the cost assumptions, validate the empirical proxy more directly, and add the requested cost-regime sensitivity analysis.
>
> Best regards,
>
> The Authors

---

### Review · Reviewer_XNud · 2026-04-12

**Summary Of Contributions:**

## Summary of Contributions
This paper proposes a normative and operational reframing of algorithmic fairness. Rather than treating statistical criteria such as demographic parity (DP), equalized odds (EO), and calibration as fairness itself, it argues that these metrics should be understood as surrogates for a more fundamental object: the allocation of harms across individuals with heterogeneous subjective costs. The paper formalizes this perspective by defining individual expected harm in terms of personalized false-positive and false-negative costs, assembling these into a harm-allocation vector, and then evaluating that allocation using fair-division style axioms such as envy-freeness and proportionality.

On top of this axiomatic framework, the paper makes two broader theoretical moves. First, it studies when familiar statistical fairness metrics can validly stand in for harm-based fairness, and shows that strong structural assumptions are needed for such surrogate validity. Second, it introduces a bounded-rationality perspective in which an intermediary has imperfect information about individual costs and access only to restricted policy classes, leading to an irreducible approximation floor for fairness. The experiments section is then used to stress test this framework, illustrating how surrogate metrics and harm-based diagnostics can diverge under heterogeneous costs and mediated decision constraints.

## Key Strengths
- **Conceptual ambition paired and a fairly coherent mathematical formulation:** The paper does not merely criticize existing fairness metrics in a generic way; instead, it offers a more precise account of what those metrics are implicitly trying to proxy. By moving from prediction errors, to individualized harms, to axiomatic evaluation of harm allocations, the paper gives a structured and interpretable foundation for its critique. That makes the contribution feel more principled than many papers that simply note that fairness metrics can conflict or fail to capture nuance.
- **Thoughtful integration of normative and operational viewpoints:** The fair-division framing gives the paper a clear normative core, while the bounded-rationality framing makes the discussion more realistic by recognizing that institutions do not observe true preferences or harms directly and are limited in what policies they can implement. I think this combination helps distinguish the work from papers that are only philosophical on one side or only optimization-based/fairness theory-heavy.
- **Clear organization of theoretical claims:** Unlike many pepers, he paper does not rest on a single impossibility observation, but it develops a pretty compelling layered argument involving necessary conditions for surrogate validity, counterexamples showing structural failure, reinterpretations of classical incompatibilities, and lower-bound style results under information and policy constraints. Even where some claims are partly interpretive, the overall structure gives the paper a clear internal logic that I could follow.
- **Empirical completeness**: Finally, the empirical section is valuable as a constructive stress test of the theory. It makes a genuine effort to operationalize the paper’s central distinction between metric-space fairness and harm-space fairness. I also noticed sufficient breadth across multiple tasks, models, and intervention families.

## Key Weaknesses
- **Novelty claims:** The framing is interesting, but parts of the introduction risk suggesting a sharper break from prior work than is fully justified. There is already meaningful literature on welfare-aware, preference-aware, and justice-oriented fairness, and the paper is strongest when presented as a synthesis and sharpening of those ideas rather than as the first move away from metric-centric thinking.
- **Better justification of normative choices:** In particular, the choice to foreground envy-freeness and proportionality as the main harm-based axioms is intuitive, but the paper would benefit from a clearer explanation of why these are the most appropriate or useful first principles in this setting relative to other plausible fairness or welfare criteria. Relatedly, the interpretation of cross-person comparisons through subjective costs is promising, but it sits on a delicate normative foundation that deserves slightly more explicit discussion.
- **Framing of experimental findings:** The experiments are clearly motivated and useful, but they necessarily rely on synthetic cost models and proxy diagnostics for the theoretical objects introduced earlier. The most important case is the cost-stratum envy measure, which is a sensible practical approximation but not identical to the underlying axiomatic object. Because of that, the empirical section is strongest as evidence that the theory matters in practice, rather than as definitive validation of stable empirical profiles of particular fairness notions or algorithm families.

**Audience:**

Yes

**Audience Explanation:**

I believe this work will be broadly of interest to the TMLR audience. It provides a principled framework for asking what standard fairness metrics are actually measuring, and when they fail to capture the distribution of harms that models impose on individuals. More broadly, it offers a way to connect statistical evaluation, normative reasoning, and implementation constraints in a single mathematical framework, which is valuable not only for fairness research but also for any ML setting where proxy objectives may diverge from the outcomes practitioners ultimately care about.

**Broader Impact Concerns:**

I believe the broader-impact statement is already pretty solid. It covers the two biggest risks well: (1) misuse as an anti-fairness argument and (2) value laundering through convenient cost assumptions; it also notes privacy, representational bias, and stakeholder burden when eliciting costs. The calls to action as robustness checks, uncertainty sets, documented elicitation, and human oversight, all seem appropriate mitigations.

The only thing I would flag is institutional power over whose values get encoded (especially given the underlying focus on heterogeneous, noisy cost estimates; one sentence here is sufficient). A smaller omission is that it could note the risk of false precision: once harms are quantified, decision-makers may over-trust the resulting numbers and understate the moral contestability of those estimates.

**Claims And Evidence:**

Yes

**Claims Explanation:**

The paper is **conceptually and mathematically sound**, and its strongest contribution is the disciplined formal chain it builds from a decision rule to a normative fairness evaluation. In particular, it starts with a policy $\pi$, moves to individualized error exposure, defines individualized harm as
$$
H_i(\pi)=c_i^{FP}\Pr_\pi(FP\mid i)+c_i^{FN}\Pr_\pi(FN\mid i),
$$
assembles these into a harm-allocation vector $H(\pi)$, and then evaluates that vector using fair-division-style axioms. This progression is one of the paper’s main strengths, because it makes the later theoretical claims feel grounded in a precise mathematical object rather than in a vague philosophical intuition.

A second major strength is the bounded-rationality formalization. The objective
$$
\mathrm{Ach}(\hat c,\varepsilon)=\inf_{\pi\in\Pi_{\text{simple}}}\sup_{c\in C(\hat c,\varepsilon)}\mathrm{Viol}(\pi;c)
$$
is well motivated and gives the paper an operational backbone. It meaningfully captures the fact that institutions do not observe subjective harms directly and do not act with unrestricted policy classes. This is not just a philosophical add-on; it is a serious mathematical extension of the fairness problem.

### Theory
The main theoretical results are **credible, meaningful, and appropriately structured**. Theorem 7.1 makes the central point that statistical constraints such as DP, EO, and calibration cannot uniformly control harm-based fairness unless subjective costs are restricted enough to collapse into a comparable unit. The proof idea is straightforward and convincing: because the metric is blind to costs, one can scale a single individual’s valuation and increase harm-based unfairness without changing the statistical constraint.
- Theorem 7.2 is also sound as a worst-case result. It shows that a policy can satisfy a statistical fairness constraint while inducing arbitrarily large harm-based violations. The theorem’s force is not that real harms are literally unbounded, but that **uniform surrogate validity fails** without structural assumptions on subjective value. That is a strong and important theoretical message.
- The lower-bound result in Section 7.4 is smaller in scope but especially elegant. It cleanly shows that once imperfect value information and policy restrictions are introduced, a nonzero residual fairness gap can be unavoidable. I view this as a successful mechanism theorem: it isolates a simple source of irreducible approximation error rather than overreaching.
- One place where the theory is slightly less complete is that some of the results are more **necessary-condition or impossibility style** than constructive. The paper is very strong at showing when surrogate validity fails, but somewhat less explicit about which concrete restricted regimes would recover partial validity. That does not undermine the theory, but it does mean the paper is somewhat stronger on critique than on delineating the positive boundary of when surrogate metrics can still be normatively meaningful.

### Normative framework

The paper’s normative framing is **serious and well structured**, not superficial. The move from rate-based fairness to harm allocation under heterogeneous subjective value is persuasive, and the paper does a good job motivating why fairness metrics should be treated as surrogates rather than as fairness itself.

The use of envy-freeness and proportionality is also mathematically coherent. The cross-evaluated harm quantity
$$
H_{i\leftarrow j}(\pi)=c_i^{FP}\Pr_\pi(FP\mid j)+c_i^{FN}\Pr_\pi(FN\mid j)
$$
gives the paper a rigorous way to define interpersonal comparisons under heterogeneous valuations. From there, the paper defines envy-freeness through
$$
H_{i\leftarrow i}(\pi)\le H_{i\leftarrow j}(\pi)\quad\forall i,j
$$
and proportionality through
$$
H_{i\leftarrow i}(\pi)\le \frac1n\sum_{j=1}^n H_{i\leftarrow j}(\pi)\quad\forall i.
$$
This is a thoughtful adaptation of fair-division ideas to the allocation of errors and harms.

At the same time, the normative layer is the least fully settled part of the paper. The framework is coherent, but not exhaustive. In particular, the paper does not fully resolve why envy-freeness and proportionality should be the primary axioms rather than, for example, maximin, leximin, or other inequality-sensitive welfare criteria. Likewise, the status of the subjective costs $c_i^{FP},c_i^{FN}$ remains somewhat open: are they preferences, welfare losses, elicited burdens, or normative modeling devices? None of this makes the framework unsound, but it does mean the paper is best understood as a **well-developed normative proposal** rather than a definitive account of fairness.

### Empirical section

The empirical section is **sound as a stress test of the theory**. The experiments ask the right questions: whether surrogate metrics and harm-based diagnostics diverge under controlled cost heterogeneity, whether noisy or coarse value information induces residual unfairness, and whether those patterns persist across tasks, models, and intervention families. In that sense, the empirical section is well aligned with the theoretical core.
- The main empirical issue is one of **construct validity**, not unreliability or lack of care. The experiments necessarily rely on measurable proxies for the theoretical objects introduced earlier, most notably cost-stratum envy rather than full individual-level envy. The paper is commendably explicit about this, and that transparency is a genuine strength. Still, because much of the empirical narrative relies on these diagnostics, the results are strongest as evidence that the theory’s distinctions are **empirically detectable under principled operationalizations**, rather than as direct measurement of the full theoretical fairness notions.
- I also think the empirical results support a narrower but still important claim: surrogate–axiom misalignment can recur in practice under the proposed harm model. They are somewhat less naturally read as establishing a stable empirical taxonomy of methods such as DP, EO, calibration, or broader fairness interventions. The experiments show a recurring and practically relevant phenomenon, but not necessarily a universal ranking of fairness methods in harm space.

### Overall soundness verdict
Overall, I think the paper is sound. The conceptual setup is coherent, the mathematics is meaningful and internally consistent, the theory says something nontrivial, and the experiments genuinely engage the paper’s central claims. The main limitations are not that the paper is unsound, but that some parts of the normative interpretation and empirical operationalization remain somewhat more provisional than the formal core.

**Requested Changes:**

### 1. Strengthen positioning
The paper should position itself more explicitly as a synthesis across several literatures rather than as a wholesale departure from prior fairness work. Its contribution is strongest when framed as combining welfare-/utility-aware fairness, axiomatic fair division, and fairness under mediation or bounded rationality into a single framework for understanding statistical metrics as surrogates for harm allocation.

The related-work section would be materially stronger if it incorporated the following papers I found that are more related to the paper's direct positioning:
- **Heidari, Loi, Gummadi, and Krause (2019)**, *A Moral Framework for Understanding Fair ML through Economic Models of Equality of Opportunity*
- **Binns (2018)**, *Fairness in Machine Learning: Lessons from Political Philosophy*
- **Khan and Hanna (2022)**, *Towards Substantive Conceptions of Algorithmic Fairness*
- **Menon and Williamson (2018)**, *The Cost of Fairness in Binary Classification*
- **Corbett-Davies and Goel (2018)**, *The Measure and Mismeasure of Fairness*
- **Long (2020/2021)**, *Against False Positive Rate Equality as a Measure of Fairness*
- **von Kügelgen / Karimi et al. (2022)**, *On the Fairness of Causal Algorithmic Recourse*
- **Bhaskar, Sricharan, and Vaish (2021)**, *On Approximate Envy-Freeness for Indivisible Chores and Mixed Resources*

I think that these citations would help the authors make some points more precisely: 1. that fairness metrics already have recognized normative content; 2. that individualized error costs place the paper close to cost-sensitive and decision-theoretic fairness; and 3. that the paper’s “allocation of harms” framing fits especially naturally into the literature on bads and chores, not just generic fair division.

The paper should also say more directly how its framework relates to **individual fairness**. Since one of its main motivations is that group-level statistics miss morally relevant individual heterogeneity, the paper should explain whether it is replacing, extending, or complementing similarity-based individual fairness.

### 2. Clarify the modeling choices

I think the paper should clarify more explicitly what the subjective costs $c_i^{FP}$ and $c_i^{FN}$ are meant to represent. At present, they can be read in several ways: as preferences, welfare losses, normative severity weights, or abstract modeling devices. The framework does not break under this ambiguity, but the interpretation of the results changes depending on which of these readings is intended.

I would also encourage the authors to add one sentence explaining that the linear harm model

$$ H_i(\pi)=c_i^{FP}\Pr_\pi(FP\mid i)+c_i^{FN}\Pr_\pi(FN\mid i)$$

is a deliberate minimal abstraction, chosen for interpretability and tractability, rather than a claim that all harms are fully captured by additive FP/FN costs.  A smaller but worthwhile clarification is whether the fairness axioms are intended to apply primarily to a finite realized population, an empirical sample, or a population-level object induced by a distribution (I was a bit confused on my first read). Because the paper alternates between indexed individuals and distributional notation, making this explicit would improve precision.

### 3. Tighten the normative interpretation

The paper would benefit from a slightly fuller justification for why **envy-freeness** and **proportionality** are the main axioms. The current rationale is already good: envy-freeness captures comparative grievance, while proportionality captures concentration of burden. But since these axioms carry much of the paper’s normative weight, I think the authors should say more clearly why these are especially appropriate here relative to other plausible principles such as maximin, leximin, or inequality-sensitive welfare criteria.

Relatedly, the paper should clarify that the cross-evaluation quantity

$$ H_{i\leftarrow j}(\pi)=c_i^{FP}\Pr_\pi(FP\mid j)+c_i^{FN}\Pr_\pi(FN\mid j)$$

is being used as a **normative comparison device**. That would help readers see that the paper is not trying to model literal preference transfer across individuals, but rather a complaint-based way of evaluating whether one person has reason to object to another’s allocation.

I would also keep emphasizing that proportionality is being used here as a cap-style protection against concentrated harm, not as the unique or complete fairness principle. That point is already present, but it is important enough to deserve more prominence in the main exposition.

### 4. Strengthen empirical framing
The most important empirical revision is to sharpen the **construct-validity bridge**. The experiments operationalize the theory through measurable diagnostics, especially cost-stratum envy, but they do not directly observe the full theoretical fairness objects introduced earlier. The paper is already commendably transparent about this, and I think it should lean into that transparency rather than leave it understated. I would also encourage the authors to keep the empirical message focused on the strongest justified claim: surrogate–axiom misalignment is **recurring and practically visible** under the paper’s harm model. The experiments do support that. They do not need to be read as establishing a universal empirical taxonomy of fairness methods in harm space.

---

> ### Author Response · Authors · 2026-05-01
> **Response to Reviewer XNud**
>
> Dear Reviewer XNud,
>
> Thank you very much for your careful and generous review. We especially appreciate your reading of the paper as a framework connecting statistical fairness, individualized harm, fair-division axioms, and bounded rationality. Your comments helped us refine the manuscript’s positioning, clarify the subjective-cost model, and state the theoretical and empirical claims more cautiously.
>
> We revised the Introduction and Related Work to position the paper less as a departure from prior fairness research and more as a synthesis and sharpening of several related lines of work. The revised manuscript now explicitly connects our framework to statistical fairness, welfare- and utility-based fairness, preference-aware fairness, axiomatic fair division, causal recourse, and fairness under mediation or uncertainty. We also added discussion of the normative content already carried by statistical fairness metrics, so that DP, EO, and Calibration are treated as meaningful surrogate constraints whose harm-based interpretation must be made explicit.
>
> We also clarified the relation to similarity-based individual fairness. The Introduction now states that our approach is complementary rather than substitutive: similarity-based individual fairness asks whether similar individuals receive similar treatment, while our framework asks how error-induced burdens are distributed when individuals attach different costs to false positives and false negatives.
>
> Your comments on the cost model were particularly helpful. Section 4 now defines $c_i^{\mathrm{FP}}$ and $c_i^{\mathrm{FN}}$ as harm weights, not as directly observed preferences or complete welfare quantities. We explain that these weights may come from stakeholder elicitation, welfare-loss estimates, legal or policy principles, expert assessment, or stakeholder-informed severity judgments. We also clarify that the linear FP/FN harm model is a tractable first-order representation of error-induced burden, not a claim that all relevant harms reduce to two linear terms.
>
> We revised the formal setup to avoid ambiguity about the evaluation object. Section 3 now distinguishes distribution-level quantities, finite evaluation populations, and empirical test-sample instantiations. It also adds a realized-label versus expected-label accounting distinction, allowing the framework to be read either as ex post evaluation of realized FP/FN burdens or as ex ante accounting based on conditional label probabilities.
>
> The fair-division section was revised to make the normative role of the axioms clearer. Section 5 now presents envy-freeness as a diagnostic of comparative grievance and proportionality-style protection as a share-based protection against burden concentration. We also clarify that $H_{i \leftarrow j}$ is a cross-evaluation device: individual i evaluates individual j’s assigned error-event burden using i’s own harm weights. It is not meant to represent a literal transfer of another person’s situation.
>
> In response to your request for positive boundary conditions, Section 7.1 now includes a surrogate-alignment result. The revised paper therefore does not claim that DP, EO, or Calibration are useless or always misleading. Instead, it identifies conditions under which statistical criteria can track harm-based comparisons, and then shows how this relationship can break under cost heterogeneity.
>
> We also revised the empirical framing. Section 8 now presents the experiments as theorem-guided stress tests rather than as a universal ranking of fairness criteria. Appendix E.3 adds a new validation analysis comparing the cost-stratum envy proxy with exact individual-level envy on smaller subsets, and Appendix E.4 adds a new cost-regime sensitivity analysis varying both between-group cost scaling and within-group FP/FN heterogeneity. These analyses support the use of the proxy as a scalable comparative diagnostic while avoiding the stronger claim that it exactly reproduces all pairwise envy relations.
>
> Finally, we revised the Broader Impact Statement to include the concerns you raised about institutional power and false precision. The revised text emphasizes that the actors who define or operationalize harm costs may differ from those who bear the harms, and that numerical harm scores can make uncertain and contestable value judgments appear more objective than they are.
>
> Thank you again for these constructive suggestions. They helped us make the paper more precise in its positioning, modeling assumptions, and empirical interpretation.
>
> Best regards,
>
> The Authors

---

### Review · Reviewer_4LGy · 2026-04-28

**Summary Of Contributions:**

This work makes the observation that when a machine learning prediction is used by some policy to make decisions or take actions on individuals, the harm incurred by mistakes in that decision-making process may be felt heterogeneously. For instance, if I do not receive a 5 dollar microloan, despite the fact that I could pay it back later, it will not effect my life particularly. However, another individual in a country where the purchasing power of 5 dollars could give them enough capital to start up a business that they otherwise cannot will be substantially harmed by the policy's mistake.

In this paper, the authors advocate that such an individual's expected harm from an allocative process, where an individual's harm incurred is based on their individual costs of incurring a false negative or false positive and the probabilities of such events, is a better foundation for considering the fairness of such policies. They suggest envy-freeness and proportionality based notions, and demonstrate the tradeoffs inherent to such a framing as one might expect. The authors model the estimation problem of determining costs incurred by individuals two ways---by assuming access only to a discretization of ``cost strata" across individuals and through pointwise (i.e. individual-wise) bounds on the deviation between an individual's true cost and the estimated cost that the intermediary agent who is deciding on what policy to use to map from scores to actions has access to.

**Audience:**

Yes

**Audience Explanation:**

I don't know of any work that explicitly considers heterogeneous welfare/harm incurred by algorithmic decision making in the way that the authors here do.

**Claims And Evidence:**

Yes

**Claims Explanation:**

"The proofs are clear and correct, requiring only elementary arguments. The appendix, which runs ~twenty additional pages, consists largely of restatements and minor variations of results already established in the main text."

**Requested Changes:**

A. [Critical] Incorporation of the underlying label uncertainties.

The authors chose to define the harm incurred by a policy on individual i in terms of false negatives and positives. These false positives and negatives are written in terms of the realized label of individual i, as opposed to their expected label ($\mathbb{E}[y \vert x]$). This is partially because harm is defined only in-sample, for each of the collection of $n$ individuals under consideration. However, I think it is important to consider, because it isn't clear how to interpret the paper's findings when we imagine that some individuals might, from the perspective of the policy or score functions, be entirely indistinguishable from one another---and thus holding policy-makers to a standard that is in terms of the individuals' realized labels as opposed to conditional label probability is too strong of a demand.

Consider the following example, where there are two groups of people, those in group $A$, and those in group $B$, who experience costs differently. In particular, people in group A have a cost 2 of false positives, and people in group $B$ have a cost 2 of false negatives. Let's say our population is evenly split between group A and B, and that individuals in group A all have label 0 while individuals in group B all have label 1. However, from the features available to the scoring model s and policy pi, we'll say that we cannot distinguish between the groups---in other words, the best the scoring model and policy pi can do is to flip a coin for every individual in order to make a decision about them. With this randomized policy, which, again, is the best possible/Bayes optimal policy and which no (generalizable) policy can compete with, what happens?

For any individual $i$ in group A, their true label is 0, but half the time the policy will label them 1. So $H_i$ is 2(1/2)=1. Note that the probability of a false positive for any point in group B is 0, since none of the points in B have true label 0. What this means is that if $i$ compares themself to $j$, they will envy them: $H_{i \leftarrow j} = 2(0)+0(1/2)=0$. The same thing occurs symmetrically in the other direction.

In other words, the optimal policy incurs maximum envy, across the entire population of individuals, and we should never expect to do better than this unless we can learn additional features that can elicit which of groups $A$ and $B$ a datapoint is a member of.

This example is a toy one, but it points to a broader question: how should we interpret harm that is due to aleatoric label uncertainty? In some contexts, it seems like avoiding such harms might be asking for far too high a bar for a policy-maker, akin to requiring them to perfectly predict the outcome of a coin toss. In other contexts, such as when the uncertainty is due to a failure of the data collection process and there were in fact ways in which the uncertainty could have been mitigated but which were not chosen, it might be reasonable to request that this harm is avoided. Regardless, the paper should engage with this question.

B. [Critical] Assumptions on the cost estimation.

A major weakness of this paper is that costs are assumed to be approximately correct pointwise. This is an extremely strong assumption, and does not match my intuition that such estimates would only at best be correct in expectation over some population, rather than individually. E.g. we might be able to, using survey data etc., bucket individuals with demographically similar ones and say something about the expected cost of a false negative or positive for that particular group, but it seems like an extremely hard problem from a learning theory perspective to ever be able to elicit true individual costs. In a sense this leads to a "turtles all the way down" problem: the reason why we are interested in learning these costs is because it is reasonable to believe that there is heterogeneity in how individuals experience harms, but then this same kind of heterogeneity could also mean that our ability to estimate costs well for different individuals may differ substantially across the population. And if we cannot ever compute individual costs, we also cannot compute individually incurred envy. One thing this means is that the lower bounds on harm-based fairness are probably substantial underestimates (as well as due to what I discussed above), and in a sense should be considered best-case bounds.

C. [Critical] Concision.

This paper's main contribution is a nice and simple idea: that group-based notions of fairness do not adequately capture heterogeneity in how harm is felt, and that thus more effort should be made in the model and policy design pipeline to measure individual-level harms and to include this consideration as part of the process of choosing a best policy. However, it risks obfuscating the message due to its 50 pages of rewording this observation. The entirety of section 9 could be removed; the additional proofs in the appendix do not particularly add to the contribution (they are just a slight reformulation and formalization of the proofs in the main body, which are sufficiently simple in terms of mathematical machinery that such expansion is unnecessary).

C. [Noncritical] Detangling harms due to structure of allocation and harms due to the mistakes a model makes.

One tricky aspect of allocation of goods (and bads) is that it is not entirely clear to me that it is always the case that harm or benefit is only incurred due to a model's mistakes. For instance, in the context of allocating food to hungry people or housing refugees, there is a sense in which failing to allocate the good or opportunity to someone harms them---or at least certainly does not help them---regardless of whether or not, in terms of ground truth, they were the ``correct" individual to receive the good or opportunity. However, this paper positions harms as only being in terms of model mistakes, and thus seems to miss this aspect of harms in allocative settings.

---

> ### Author Response · Authors · 2026-05-01
> **Response to Reviewer 4LGy**
>
> Dear Reviewer 4LGy,
>
> Thank you very much for your careful and constructive review. We appreciate your positive assessment of the paper’s main idea and theoretical correctness, and we found your critical comments especially helpful for clarifying the scope of the framework and reducing unnecessary expansion.
>
> Your concern about label uncertainty led us to revise the formal setup in Section 3 and the limitations discussion in Section 9.6. The revised manuscript now distinguishes realized-label accounting from expected-label accounting. The main experiments still use realized FP/FN burdens, matching standard empirical evaluation practice, but Section 3 now also defines the corresponding expected-label primitives using conditional label probabilities. Section 9.6 further clarifies that some realized error burdens may reflect avoidable model mistakes, while others may arise from aleatoric or label uncertainty that even a Bayes-optimal policy cannot eliminate. We now state explicitly that whether such irreducible uncertainty should count as a fairness violation depends on the normative target of evaluation.
>
> We also revised the paper’s treatment of individual cost information. In Sections 4 and 6, $c_i^{\mathrm{FP}}$ and $c_i^{\mathrm{FN}}$ are now described as harm weights or contestable cost representations, not as perfectly observed individual truths. We clarify that these weights may come from elicitation, welfare-loss estimates, legal or policy principles, expert assessment, or stakeholder-informed severity judgments. Section 6 now emphasizes that institutions typically operate with noisy, coarse, or proxy-based cost representations and restricted policy classes. The bounded-rationality results should therefore be read as favorable-information or benchmark analyses: if residual violation remains even under structured proxy information and simple uncertainty sets, then real institutional settings may face still larger difficulties.
>
> Your point about concision was also important. We substantially shortened and reorganized the discussion and appendix. Section 9 now focuses on the interpretation needed for the revised paper: conditional surrogate alignment, bounded-rationality floors, distinct harm profiles, reporting guidance, and limitations. In the appendix, we kept compact proofs and scope clarifications, while avoiding unnecessary repetition of arguments already made in the main text. The remaining appendix material serves two targeted purposes: supporting the main theoretical claims and reporting the additional empirical validation requested by the reviewers.
>
> We also clarified the scope of the harm model. The revised manuscript now states in Sections 3 and 9.6 that the formal analysis focuses on error-induced burdens represented by false-positive and false-negative events. We do not claim to cover all harms that can arise from allocation systems, such as the loss of a scarce good even when a prediction is technically correct. Appendix C and Appendix F explain how the same logic could extend to richer policy-dependent harm bundles, but the main claims are now explicitly limited to the error-induced setting studied in the paper.
>
> Thank you again for these helpful comments. They led us to make the paper more precise, shorter, and clearer about what the framework does and does not claim.
>
> Best regards,
>
> The Authors

---

### Decision · Action_Editor_Y32t · 2026-06-01

**Recommendation:** Reject

**Additional Comments:**

In addition to the technical suggestions, I encourage the authors to carefully address the reviewers' concerns about the presentation and writing.

**Audience:**

Yes

**Audience Explanation:**

The paper proposes a harm-allocation framework for algorithmic fairness, arguing that statistical criteria such as demographic parity, equalized odds, and calibration serve as surrogates for the distribution of error-induced burdens across individuals under heterogeneous subjective costs. The authors formalize this using individual FP/FN weights and fair-division results, including in the presence of noisy and coarse cost information.

The reviewers agree that the topic is timely and relevant for the TMLR audience.

**Claims And Evidence:**

No

**Claims Explanation:**

The paper proposes a harm-allocation framework for algorithmic fairness, arguing that statistical criteria such as demographic parity, equalized odds, and calibration serve as surrogates for the distribution of error-induced burdens across individuals under heterogeneous subjective costs. The authors formalize this using individual FP/FN weights and fair-division results, including in the presence of noisy and coarse cost information.

The reviewers agree the topic is timely and relevant for the TMLR audience, and the authors addressed several comments in revision, including clarifying that individual FP/FN costs are harm weights rather than observed ground-truth preferences, discussing proxy-based cost information, and adding a cost-regime sensitivity analysis and a validation of the cost-stratum envy proxy.

The reviewers were split. One found the revision satisfactory and leaned towards acceptance; the other two felt their central concerns were not fully addressed. The main concern was that the framework does not adequately handle label uncertainty; reviewers questioned whether individualized FP/FN costs can be realistically specified rather than only approximated; and several reviewers found the manuscript long, repetitive, and in need of substantial tightening. I agree that addressing these points is critical to the paper's central contributions.

The paper is promising and likely to be of interest to the TMLR audience. I encourage a future submission that addresses these concerns.

**Resubmission Of Major Revision:**

The authors may consider submitting a major revision at a later time.